# Stromal fibroblasts induce metastatic tumor cell clusters via epithelial–mesenchymal plasticity

Yuko Matsumura[1,2], Yasuhiko Ito[1], Yoshihiro Mezawa[1], Kaidiliayi Sulidan[1,2], Yataro Daigo[12,13], Toru Hiraga[7] , Kaoru Mogushi[8], Nadila Wali[1], Hiromu Suzuki[15], Takumi Itoh[1] , Yohei Miyagi[9], Tomoyuki Yokose[10] , Satoru Shimizu[11], Atsushi Takano[12,13], Yasuhisa Terao[2], Harumi Saeki[1], Masayuki Ozawa[14], Masaaki Abe[1], Satoru Takeda[2], Ko Okumura[4,6], Sonoko Habu[4,6], Okio Hino[1], Kazuyoshi Takeda[5,6] , Michiaki Hamada[3] , Akira Orimo[1,16]

**Emerging evidence supports the hypothesis that multicellular tumor clusters invade and seed metastasis. However, whether tumor-associated stroma induces epithelial–mesenchymal plasticity in tumor cell clusters, to promote invasion and metastasis, remains unknown. We demonstrate herein that carcinoma-associated fibroblasts (CAFs) frequently present in tumor stroma drive the formation of tumor cell clusters composed of two distinct cancer cell populations, one in a highly epithelial (E-cadherin$^{hi}$ZEB1$^{lo/neg}$: E$^{hi}$) state and another in a hybrid epithelial/mesenchymal (E-cadherin$^{lo}$ZEB1$^{hi}$: E/M) state. The E$^{hi}$ cells highly express oncogenic cell–cell adhesion molecules, such as carcinoembryonic antigen-related cell adhesion molecule 5 (CEACAM5) and CEACAM6 that associate with E-cadherin, resulting in increased tumor cell cluster formation and metastatic seeding. The E/M cells also retain associations with E$^{hi}$ cells, which follow the E/M cells leading to collective invasion. CAF-produced stromal cell-derived factor 1 and transforming growth factor-$\beta$ confer the E$^{hi}$ and E/M states as well as invasive and metastatic traits via Src activation in apposed human breast tumor cells. Taken together, these findings indicate that invasive and metastatic tumor cell clusters are induced by CAFs via epithelial–mesenchymal plasticity.**

## Introduction

The complete epithelial–mesenchymal transition (EMT) program, a main driver of the invasion-metastasis cascade, results in carcinoma cells losing all epithelial traits as well as cell–cell adhesion, instead acquiring mesenchymal properties (Thiery et al, 2009). EMT induces motile, invasive, and tumor-initiating abilities in carcinoma cells, facilitating their intravasation as single cells into the bloodstream and colonization of distant organs via subsequent induction of mesenchymal–epithelial transition (MET) (Thiery et al, 2009; Lambert et al, 2017).

In contrast to the single cell dissemination via EMT, groups of epithelial carcinoma cells maintaining their cell–cell adhesion migrate collectively as clusters in culture (Friedl et al, 2012; Mayor & Etienne-Manneville, 2016). Leader cells as evidenced by the mesenchymal trait are located at the front of the follower epithelial cancer clusters and drive their collective migration in response to environmental cues (Westcott et al, 2015; Mayor & Etienne-Manneville, 2016). Tumor cell clusters, designated "tumor budding" and "tumor emboli," are also assumed to intravasate by collective migration or passive shedding into the circulation in cancer patients (Grigore et al, 2016). Furthermore, circulating tumor cell (CTC) clusters seed metastases significantly more often than single cancer cells in experimental murine models (Maddipati & Stanger, 2015; Cheung et al, 2016) and breast cancer patients (Aceto et al, 2014).

Unlike complete EMT, "epithelial–mesenchymal plasticity" is controlled by multiple variants of the core EMT program that produces carcinoma cells with partial EMT, thereby generating hybrid cells expressing both epithelial and mesenchymal traits (Ye & Weinberg, 2015; Nieto et al, 2016; Lambert et al, 2017; Brabletz et al, 2018). These cells are more frequently detected in various human carcinomas than are those with complete EMT (Yu et al, 2013;

[1]Department of Molecular Pathogenesis, Graduate School of Medicine, Juntendo University, Tokyo, Japan   [2]Department of Obstetrics and Gynecology, Graduate School of Medicine, Juntendo University, Tokyo, Japan   [3]Department of Electrical Engineering and Bioscience, Faculty of Science and Engineering, Waseda University, Tokyo, Japan   [4]Atopy Research Center, Biomedical Research Center, Graduate School of Medicine, Juntendo University, Tokyo, Japan   [5]Division of Cell Biology, Biomedical Research Center, Graduate School of Medicine, Juntendo University, Tokyo, Japan   [6]Department of Biofunctional Microbiota, Graduate School of Medicine, Juntendo University, Tokyo, Japan   [7]Department of Histology and Cell Biology, Matsumoto Dental University, Nagano, Japan   [8]Intractable Disease Research Center, Graduate School of Medicine, Juntendo University, Tokyo, Japan   [9]Molecular Pathology and Genetics Division, Kanagawa Cancer Center Research Institute, Yokohama, Japan   [10]Department of Pathology, Kanagawa Cancer Center, Yokohama, Japan   [11]Department of Breast and Endocrine Surgery, Kanagawa Cancer Center, Yokohama, Japan   [12]Center for Antibody and Vaccine Therapy, Research Hospital, Institute of Medical Science, The University of Tokyo, Tokyo, Japan   [13]Department of Medical Oncology and Cancer Center, Shiga University of Medical Science, Otsu, Japan   [14]Graduate School of Medical and Dental Sciences, Kagoshima University, Kagoshima, Japan   [15]Department of Molecular Biology, School of Medicine, Sapporo Medical University, Hokkaido, Japan   [16]Cancer Research (CR)-UK Stromal-Tumor Interaction Group, Paterson Institute for Cancer Research, The University of Manchester, Manchester, UK

Correspondence: aorimo@juntendo.ac.jp

Bronsert et al, 2014). Cancer cells with partial EMT also form cohesive multicellular clusters via maintenance of membrane E-cadherin (E-cad) expression to collectively invade and disseminate (Campbell & Casanova, 2016; Grigore et al, 2016; Aiello et al, 2018; Li et al, 2019). However, the roles of epithelial–mesenchymal plasticity in invading tumor cell cluster formation have yet to be investigated in detail.

Carcinoma-associated fibroblasts (CAFs), activated fibroblast populations frequently present in stroma of different human carcinomas, are competent to promote tumor invasion and metastasis (Calon et al, 2012; Zhang et al, 2013; Mezawa & Orimo, 2016). CAFs stimulate collective cell migration of cancer cell clusters in 3D cultures (Gaggioli et al, 2007). E-cad–expressing colon carcinoma cell budding is also located with stromal myofibroblasts at the invasive front of tumors (Dimanche-Boitrel et al, 1994). However, whether CAFs influence the formation of invading tumor cell clusters via epithelial–mesenchymal plasticity has not yet been determined. We, thus, sought to elucidate the relevance of tumor cell cluster formation, collective cell invasion, and metastasis to the epithelial–mesenchymal plasticity regulated by CAFs.

# Results

## CAFs induce both highly epithelial and hybrid epithelial/mesenchymal breast cancer cell populations

Tumor cell clusters invade and seed metastasis, but whether CAF-induced epithelial–mesenchymal plasticity contributes to this process remains unclear. To assess this possibility, we developed a co-implantation tumor xenograft model that allows continuous interaction between the injected tumor cells and human stromal fibroblasts within a tumor mass. Thus, we used GFP-labelled experimentally generated immortalized human mammary CAFs that acquired greatly activated myofibroblastic trait and tumor-promoting ability as compared with control mammary fibroblasts during tumor progression (Kojima et al, 2010; Polanska et al, 2011). These CAFs or control human mammary fibroblasts were also mixed with barely metastatic noninvasive breast ductal carcinoma MCF10DCIS.com (DCIS) cells (Miller et al, 2000), and these mixtures were injected subcutaneously into the flanks of immunodeficient NOD/Shi-scid IL2 γ null (NOG) mice. We also introduced a cDNA construct encoding red fluorescent protein variant (tdTomato) into DCIS cells to allow their detection in vivo.

We observed acinar structure formation with proper coverage of the p63-positive (p63$^+$) myoepithelial layer in DCIS tumors grown in the presence or absence of control fibroblasts (Figs 1A, S1A, and B). In sharp contrast, this intact acinar structure was largely lost in tumors containing CAFs, reflecting the CAF-induced invasive propensity of these tumor cells. The injected control human mammary fibroblasts and CAFs were also detected in the tumors, based on GFP fluorescence and immunostaining using human-specific antivimentin antibody (Fig S1B and C).

We stained the subcutaneous tumor sections with antibodies against E-cad, an epithelial marker, as well as ZEB1, vimentin and fibronectin, which are mesenchymal markers. Carcinoma cells admixed with CAFs showed stronger E-cad, vimentin, and fibronectin staining as well as nuclear ZEB1 staining than did those comingled with/without control fibroblasts (Figs 1A and S1D). Two distinct cancer cell populations—E-cad$^{hi}$ZEB1$^{lo/neg}$ (E$^{hi}$) cells located at the center of the tumor and E-cad$^{lo}$ZEB1$^{hi}$ (E/M) cells close to the stroma–tumor interface—were observed to be more numerous in CAF-containing tumors than in those admixed with control fibroblasts (Fig 1B). The E$^{hi}$ and E/M tumor cells were also consistently demonstrated by another set of anti–E-cad and anti-ZEB1 antibodies (Fig S1E). Such E$^{hi}$ tumor cells showed relatively strong membrane E-cad staining, whereas the E/M tumor cells exhibited more nuclear ZEB1 staining with attenuated E-cad expression (Figs 1B and S1E). Some E-cad$^+$ cancer cells also stained positive for fibronectin or vimentin in CAF-containing tumors (Fig S1F), further indicating the E/M tumor cells induced by CAFs via partial EMT.

To quantify the E$^{hi}$ and E/M cancer cell proportions, tumor xenografts arising from tdTomato$^+$ DCIS cells admixed with CAFs or control fibroblasts were dissociated into single cell suspensions before staining with anti–E-cad and anti-ZEB1 antibodies. CAF-containing tumors showed a 2.1-fold increase in E-cad$^+$ tdTomato$^+$ DCIS cell proportions as compared with those containing control fibroblasts, as gauged by flow cytometry (Fig 1C), consistent with increased E-cad staining in tumors admixed with CAFs. The E$^{hi}$ and E/M tdTomato$^+$ tumor cell proportions were also significantly increased in CAF-containing tumors relative to E-cad$^{hi}$ZEB1$^{hi}$ tumor cells (Figs 1D and S1G), reflecting the well-known inverse relationship between E-cad and ZEB1 expressions (Eger et al, 2005; Sanchez-Tillo et al, 2010). Taken together, these data further confirm CAF-induced E$^{hi}$ and E/M malignant cells within tumors.

## CAFs induce collective cell invasion of cancer cell clusters and metastasis

As CAFs have previously been demonstrated to prime collective invasion of tumor cell clusters (Gaggioli et al, 2007), we investigated their relevance to epithelial–mesenchymal plasticity. DCIS cells were seeded onto Matrigel/collagen gel embedded with CAFs or control fibroblasts. CAFs stimulated invasion of E/M cancer cells, which retained an association with the follower E$^{hi}$ tumor cell clusters significantly more than did control fibroblasts (Fig 1E), leading to collective invasion of the E$^{hi}$ and E/M tumor cell clusters. This finding mirrors a recent study describing tumor cell subpopulations, defined as leader cells, with the mesenchymal trait that can initiate collective invasion with follower cells expressing the highly epithelial feature (Konen et al, 2017; Pearson, 2019).

We next examined whether CAFs boost tumor progression in vivo. When tdTomato$^+$ DCIS cells were injected with different fibroblasts subcutaneously into the recipient mice, the CAF-containing tumors grew more rapidly than did those admixed with or without control fibroblasts (Fig S1H). To precisely evaluate CAFs' metastasis-promoting abilities, the metastasis index was used as the ratio of metastasis formation, gauged by tdTomato fluorescent intensity and nodule volume in the lungs, relative to primary tumor weight. We found that CAFs raised the lung metastasis index significantly more than did no fibroblasts and control fibroblasts (Figs 1F and S1I). However, the injected CAFs barely co-metastasized with

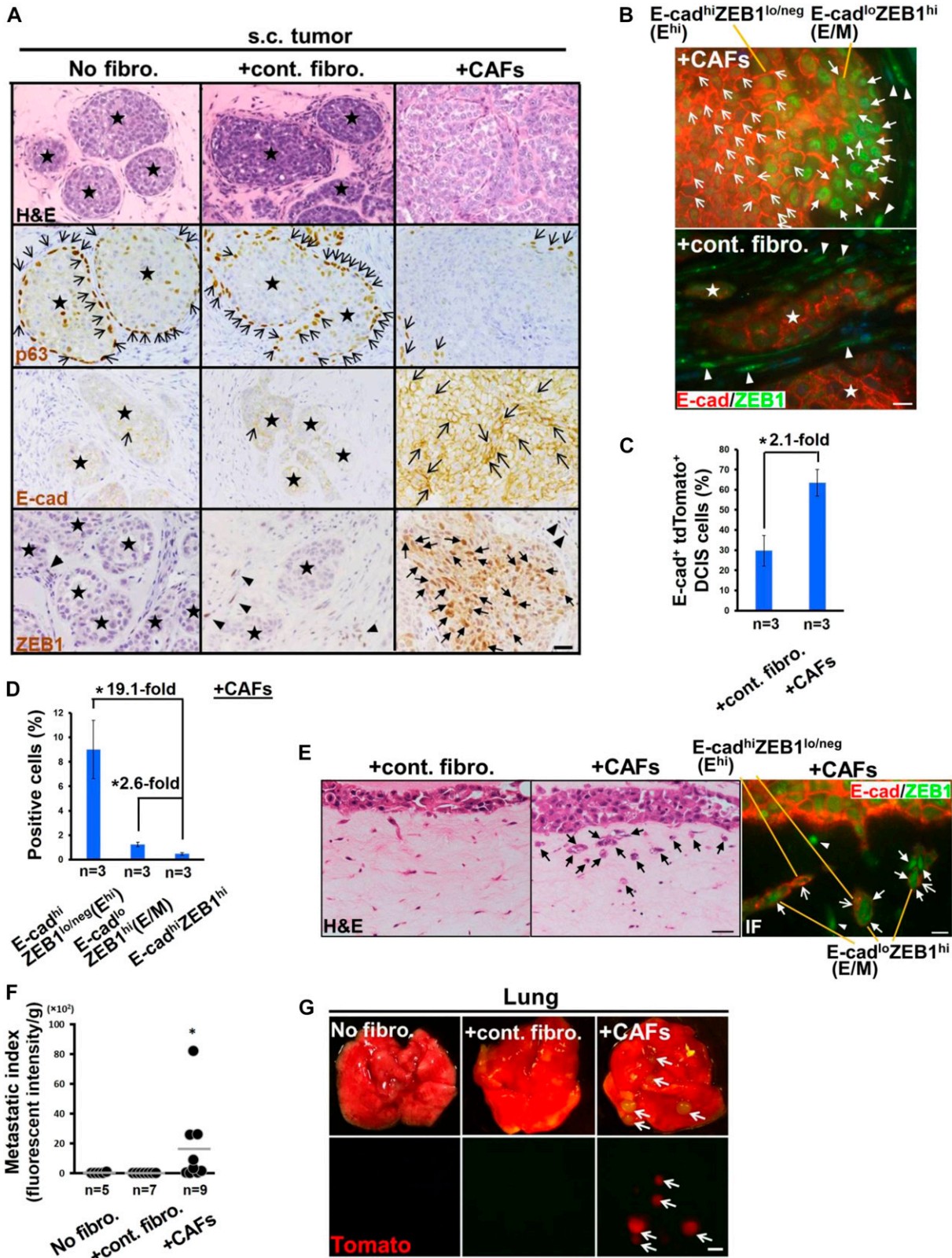

**Figure 1. CAF-induced breast carcinoma cell clusters with the E$^{hi}$ and E/M states, collective invasion, and metastasis.**
**(A)** The sections were prepared from 21-d-old DCIS tumor xenografts subcutaneously (s.c.) implanted into mice, with no fibroblasts (No fibro.), control fibroblasts (+cont. fibro.), or CAFs (+CAFs). H&E staining and immunohistochemistry using the indicated antibodies. p63$^+$ and E-cadherin (E-cad)$^+$ cancer cells (simple arrows), nuclear ZEB1$^+$ cancer cells (triangular arrows), and nuclear ZEB1$^+$ stromal cells (arrowheads) are also shown. **(B)** Immunofluorescence of paraffin sections prepared from 21-d-old tumors

carcinoma cells because of the absence of human-specific vimentin-positive stromal cells in the affected lungs (Fig S1J). tdTomato[+] liver metastasis developed in one of eight mice bearing tumors admixed with CAFs, whereas no tdTomato[+] cells were detected in other organs, including bone and brain (Fig S1K). Moreover, tdTomato[+] cells were rare in the lungs and other organs of mice bearing tumors admixed with or without control fibroblasts (Figs 1G and S1L). These findings indicate that CAFs induce the E/M tumor cells, allowing continuous association with the follower E[hi] tumor cell clusters to boost collective invasion and metastasis.

## Invasive and metastatic traits progressively conferred upon tumor cells by CAFs

Given the importance of co-evolution of tumor cells and stromal cells during tumor progression (Kojima et al, 2010), we examined whether the tumor-promoting E[hi] and E/M states are conferred upon apposed tumor cells by ongoing interactions with CAFs in a tumor mass. Thus, DCIS cells were introduced via a cDNA construct expressing both blasticidin resistance and tdTomato, before their subcutaneous injection with or without human mammary fibroblasts into recipient mice. The injected DCIS cells were then extracted from 30-d-old tumor xenografts admixed with CAFs or control fibroblasts in culture and the resulting blasticidin-resistant cells were designated DCIS[CAF1cy] or DCIS[cnt1cy], respectively (Fig 2A). DCIS cells without additions, DCIS[alone1cy], were also isolated from tumors that developed without human fibroblasts (Fig 2A). To further increase the interactions of carcinoma cells with these fibroblasts within tumors, the extracted cancer cells were again implanted with or without human fibroblasts into mice for an additional 30 d. Similarly, the blasticidin-resistant DCIS cells were extracted from tumors admixed with CAFs, with control fibroblasts, or without fibroblasts before being designated DCIS[CAF2cy], DCIS[cnt2cy], or DCIS[alone2cy], respectively (Fig 2A).

We observed that the cultured DCIS[CAF2cy] showed greater scratch wound cell invasion than did DCIS[cnt2cy] and DCIS[alone2cy] (Fig 2B). DCIS[CAF2cy], seeded onto Matrigel, also formed aggregates/spheroids that were larger and more irregular in shape with more marked E[hi] and E/M traits than did DCIS[cnt2cy] (Figs 2C and S2A).

To investigate tumor-promoting traits in vivo, different DCIS cells were subcutaneously injected with no fibroblasts into recipient mice. DCIS[CAF2cy] formed histologically invasive tumors with stronger E-cad, ZEB1, fibronectin, and vimentin staining (Figs 2D and S2B) and grew more rapidly than DCIS[alone2cy] and DCIS[cnt2cy] (Fig S2C).

Moreover, larger E[hi] and E/M cell populations were detected in tumors generated by DCIS[CAF2cy] (Fig 2E). In addition, DCIS[CAF2cy] generated lung metastatic nodules in greater number and size with a higher metastasis index than DCIS[alone2cy] and DCIS[cnt2cy] (Fig 2F) (Orimo et al, 2015). These observations are consistent with earlier findings of increased invasion and metastasis associated with the E[hi] and E/M states in cancer cells admixed with CAFs (Fig 1).

In addition, DCIS[CAF2cy], when intravenously injected into recipient mice, generated more pulmonary metastases than DCIS[alone1cy], DCIS[alone2cy], DCIS[cnt1cy], DCIS[cnt2cy], and DCIS[CAF1cy] (Figs 2G, S2D, and E) (Orimo et al, 2015), indicating that CAFs progressively confer lung-colonizing ability upon tumor cells during tumor progression. We also confirmed the CAF-promoted metastasis and E[hi] and E/M states in two other human breast cancer cell lines, MCF10CA1d (1d) (Santner et al, 2001) and MCF-7-ras cells (Orimo et al, 2005). These tumor cells, implanted with fibroblasts subcutaneously into mice, were also extracted from developing tumor xenografts. The resulting 1d[CAF1cy] and MCF-7-ras[CAF1cy] extracted from CAF-containing tumors showed greater lung-colonizing ability (Fig 2H) and higher E-cad and ZEB1 expressions (Fig S2F) than did their control 1d[cnt1cy] and MCF-7-ras[cnt1cy], respectively, extracted from those admixed with control fibroblasts.

We also investigated whether DCIS[CAF2cy] can colonize organs other than the lungs by intracardiac injection of these cells labelled with GFP into recipient mice. DCIS[CAF2cy] showed greater GFP intensity in bone and liver, common metastatic breast carcinoma sites in patients, than did DCIS[cnt2cy] and DCIS[alone2cy] (Fig 2I), indicating that CAFs promoted metastatic colonization of these organs in addition to the lungs. Collectively, our findings indicate that interactions with CAFs confer proliferating, invasive, and metastasis-forming abilities, with enhanced E[hi] and E/M states, upon apposed breast carcinoma cells during tumor progression.

## Three cell–cell adhesion molecules mediating the E[hi] state in DCIS[CAF2cy]

We next performed DNA microarray analysis and identified the CAF-induced metastatic signature (CIMS), composed of the 44 genes up-regulated in DCIS[CAF2cy] as compared with DCIS[cnt2cy] (Table S1). CIMS notably predicted poorer survival in several independent breast cancer patient cohorts (Figs 3A and S3A), indicating a clinical correlation with CAF-induced metastasis. Moreover, we found enrichment of E-cad–up-regulated genes, as gauged by gene set enrichment analysis (Fig 3B), and increased E-cad mRNA and

---

admixed with CAFs or control fibroblasts using anti–E-cadherin (E-cad) (Dako, Cat. No. IR059) and anti-ZEB1 (Cat. No. HPA027524; Sigma-Aldrich) antibodies. The presence of two distinct DCIS cell populations including E[hi] (simple arrow) and E/M (triangular arrow) tumor cells, and nuclear ZEB1[+] stromal cells (arrowheads) are shown. **(C)** Flow cytometry of the cell suspension dissociated from 30-d-old tumor xenografts raised by tdTomato[+] DCIS cells admixed with control fibroblasts or CAFs using an anti–E-cad antibody. Data represent the average of three independent experiments. **(D)** Flow cytometry of the cell suspension dissociated from 30-d-old tumor xenografts raised by tdTomato[+] DCIS cells admixed with CAFs using anti–E-cad and anti-ZEB1 antibodies. Data represent the average of three independent experiments. **(E)** H&E staining and immunofluorescence (IF) of the Matrigel/collagen gel using anti–E-cad and anti-ZEB1 antibodies. The DCIS cells seeded onto the gels embedded with control fibroblasts or CAFs invade as clusters (arrows) (H&E). The E[hi] (simple arrow) and E/M (triangular arrow) cancer cells and nuclear ZEB1[+] stromal cells (arrowheads) are also shown (IF). **(F)** Lung metastatic indices determined by fluorescent intensity at 60 d after subcutaneous injection of tdTomato[+] DCIS cells, admixed with or without the indicated fibroblasts into mice. The horizontal line represents the mean value. **(G)** Appearance of tdTomato[+] metastatic nodules (arrows) in the lungs at 60 d after subcutaneous injection, with no fibroblasts, control fibroblasts, or CAFs, into mice. Data information: Star indicates intact acinar structure of DCIS cells (A, B). Scale bars, 30 μm (A and E-H&E), 10 μm (B and E-IF), and 1 mm (G). Asterisk indicates a significant difference between the indicated groups (C, D) and relative to the No fibro. and +cont. fibro. groups (F). t Test (C, D) and Wilcoxon rank sum test (F). Error bars, SE. See also Fig S1.
Source data are available for this figure.

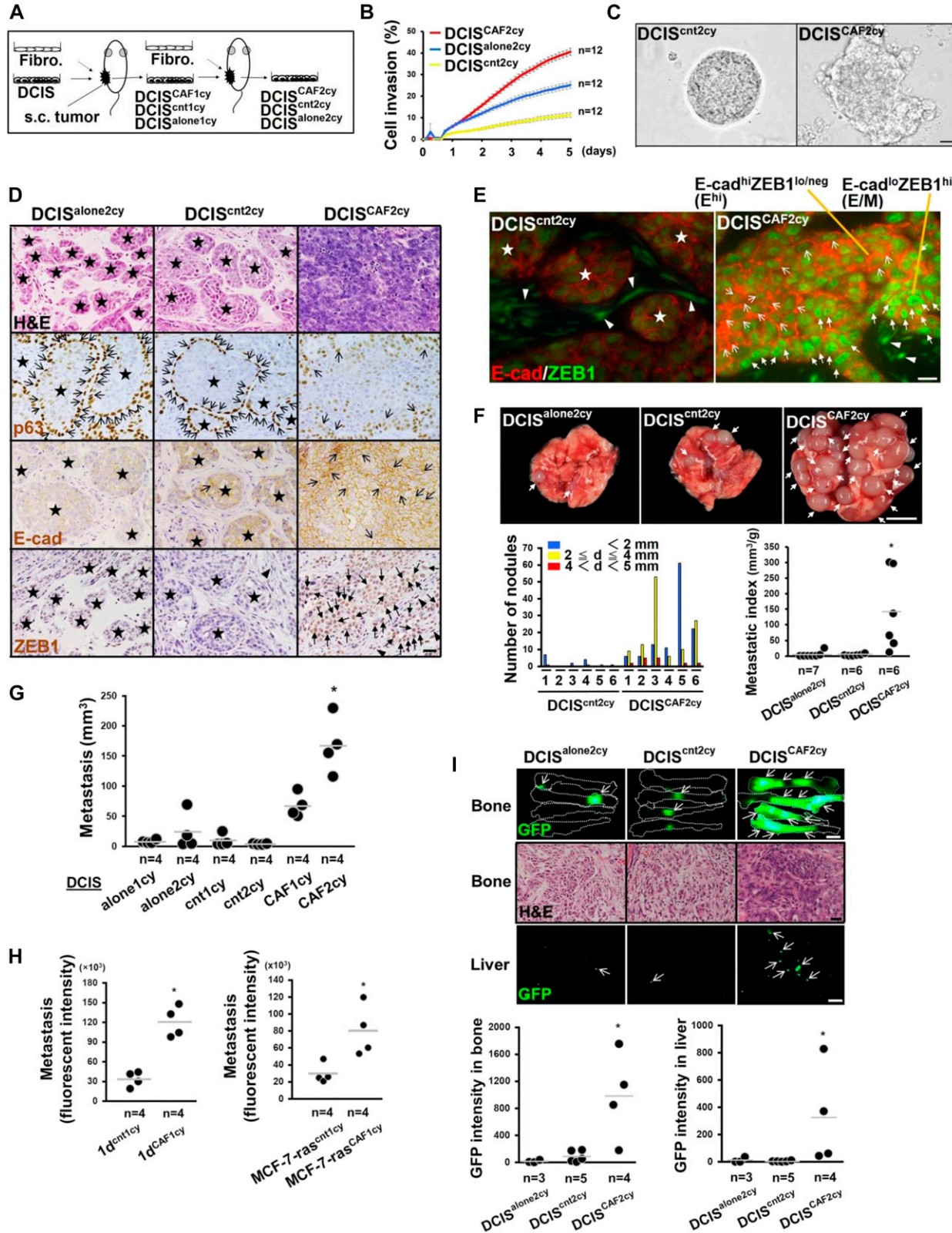

**Figure 2. Highly invasive and metastatic breast cancer cells generated by intratumoral CAFs.**
**(A)** Schematic representation of isolation of CAF-primed highly invasive and metastatic breast cancer cells. *See text for details.* **(B)** Scratch wound assay in the indicated cells. Error bars, SE. **(C)** Appearance of DCIS^cnt2cy and DCIS^CAF2cy organoids generated on Matrigel gel for 5 d. **(D)** H&E staining and immunostaining of sections prepared from 21-d-old subcutaneous tumors generated by the indicated cells using the described antibodies. p63[+] or E-cadherin (E-cad)[+] cancer cells (simple arrows), nuclear

protein expressions in DCIS[CAF2cy] (Figs 3C and S3B). In addition, DCIS[CAF2cy] markedly up-regulated expressions of carcinoembryonic antigen-related cell adhesion molecule 5 (CEACAM5; CAM5) and CEACAM6 (CAM6), tumor-promoting cell–cell adhesion molecules (Beauchemin & Arabzadeh, 2013) (Table S1). Levels of CAM5 and CAM6 mRNA and protein expressions were also significantly higher in DCIS[CAF2cy] than in DCIS[cnt2cy] and DCIS[CAF1cy] (Fig 3D and E) (Orimo et al, 2015), indicating progressive up-regulation during tumor progression. Furthermore, CAM5 and CAM6 expressions were up-regulated in the other human breast cancer cell lines, 1d[CAF1cy] and MCF-7-ras[CAF1cy], as compared with 1d[cnt1cy] and MCF-7-ras[cnt1cy], respectively (Fig S3C).

We also investigated the clinical significance of the CAF-induced E[hi] state using several different breast cancer patient cohorts. Of note, E-cad, CAM5, and CAM6 mRNA expressions mediating the E[hi] state were simultaneously expressed in human breast carcinomas (Figs 3F and S3D), indicating the CAF-induced E[hi] state to be present in breast cancer patients.

We next investigated whether E-cad, CAM5, and CAM6 interact with each other on DCIS[CAF2cy]. Double immunostaining and in situ proximity ligation assay (PLA) indeed showed co-localization and association between each pair of E-cad, CAM5, and CAM6 proteins via adherens junctions, providing important adhesive contacts on DCIS[CAF2cy] (Fig 3G). Consistently, CAM5 and CAM6 proteins reportedly show heterophilic binding to each other in addition to their homophilic binding (Oikawa et al, 1989; Zhou et al, 1993). Inhibition of E-cad, CAM5, or CAM6 expression by their corresponding shRNAs also simultaneously attenuated mRNA and protein expressions of these three genes and their associations on DCIS[CAF2cy] (Figs 3H and S3E). The interactions among the three cell–cell adhesion molecules may stabilize these proteins, in addition to their regulatory functions at the mRNA level in DCIS[CAF2cy]. These findings indicate that E-cad, CAM5, or CAM6 is crucial for maintenance of the E[hi] state of these cells.

### Tumor cell cluster formation and metastasis mediated by the E[hi] state in DCIS[CAF2cy]

Given the increased E[hi] state in DCIS[CAF2cy], we investigated how the E[hi] state influences these cells' metastatic abilities. Inhibition of E-cad, CAM5, or CAM6 expression by shRNA attenuated the lung metastases arising from DCIS[CAF2cy] via subcutaneous and intravenous injections into mice significantly more than did GFP-shRNA (Figs 4A–C and S4A–C). These findings indicate E-cad, CAM5, or CAM6 expression to be crucial for DCIS[CAF2cy] to form metastases.

Although E-cad expression was long believed to suppress tumor invasion and metastasis (Bruner & Derksen, 2018), its oncogenic roles promoting tumor cell cluster formation, collective cell migration, and metastatic colonization have recently been demonstrated (Chu et al, 2013; Shamir & Ewald, 2015). We, thus, reasoned that the E[hi] state enables DCIS[CAF2cy] to form tumor cell clusters promoting their metastatic colonization. As anticipated, DCIS[CAF2cy] showed greater cell–cell adhesion (Figs 4D and S4D), cell–cell aggregation (Fig 4E), ECM–cell adhesion (Fig 4F), and antiapoptosis (Fig 4G) than did DCIS[cnt2cy] in vitro. Inhibition of E-cad, CAM5 or CAM6 expression by shRNA attenuated these traits in DCIS[CAF2cy] significantly more than did GFP-shRNA (Fig 4D–G). To further study the roles of the E[hi] state in antiapoptosis and cell proliferation in vivo, we used experimental lung metastases produced by DCIS[CAF2cy] expressing different shRNAs before staining with anticleaved poly (ADP‑ribose) polymerase (cPARP) and anti–Ki-67 antibodies. Inhibition of E-cad, CAM5, or CAM6 expression by shRNA increased cPARP[+] apoptotic cells and decreased Ki-67[+] proliferating cells in lung metastases significantly more than did GFP-shRNA (Fig 4H). Collectively, these findings demonstrate that the E[hi] state, as exemplified by E-cad, CAM5, and CAM6 expressions, mediates tumor cell cluster formation and metastatic colonization due to increased cell–cell adhesion, aggregation, ECM–cell adhesion, antiapoptosis, and cell proliferation in DCIS[CAF2cy].

### Collective invasion and metastasis regulated by the E/M state in DCIS[CAF2cy]

As the E/M state is also increased in DCIS[CAF2cy], we reasoned that ZEB1 expression suppresses E-cad expression to generate the E/M state. To examine this possibility, we used two ZEB1-shRNAs, both of which significantly attenuated ZEB1 expression in DCIS[CAF2cy] and human mammary fibroblasts (Fig 5A). Inhibition of ZEB1 expression by shRNA further up-regulated E-cad mRNA expression in DCIS[CAF2cy] (Fig 5B), indicating E-cad expression to be down-regulated by ZEB1 in E/M cells.

As CAF-primed E/M tumor cells apparently lead to collective invasion with E[hi] tumor cells (Fig 1E), we investigated the effects of ZEB1 expression on leader E/M tumor cells and metastasis in DCIS[CAF2cy]. Inhibition of ZEB1 expression by shRNA attenuated the collective invasion of tumor cell clusters with E[hi] and E/M states (Fig 5C and D) and lung-colonizing ability (Fig 5E) significantly more than did GFP-shRNA. These findings indicate that ZEB1 expression is necessary for the E/M state to mediate collective invasion with E[hi] tumor cell clusters and metastatic dissemination in DCIS[CAF2cy].

ZEB1[+] cancer cells (triangular arrows), and nuclear ZEB1[+] stromal cells (arrowheads) are also shown. **(E)** Immunofluorescence of 21-d-old tumor sections from DCIS[CAF2cy] and DCIS[cnt2cy] using anti–E-cad and anti-ZEB1 antibodies. The Ehi (simple arrow) and E/M (triangular arrow) tumor cells, and nuclear ZEB1[+] stromal cells (arrowheads) are shown. **(F)** Representation of metastatic nodules (arrows) in whole lungs dissected from mice subcutaneously injected with the indicated cancer cells (upper). The size (d: diameter) and number of metastatic nodules are shown for the indicated groups (n = 6) (left). The lung metastatic indices were also evaluated in each group at 60 d after injection (right). **(G)** Lung metastasis volumes evaluated at 30 d after intravenous injection of the indicated cells into mice. **(H)** Metastases evaluated by tdTomato fluorescent intensity in the lungs at 30 d after intravenous injection of the indicated tdTomato[+] tumor cells into mice. **(I)** Detection of GFP fluorescence (arrow) in bone (upper) and liver (lower) of mice at 60 d after intracardiac injection of the indicated GFP[+] cancer cells. H&E staining of bone metastasis (middle) is also shown for the described groups. Quantification of GFP intensity of the indicated cells colonizing bone and liver (graphs). Data information: Star indicates intact acinar structure of DCIS cells (D, E). Asterisk indicates a significant difference relative to others (F–I). Mann–Whitney $U$ test (F, H, I) and Wilcoxon rank sum test (G). The horizontal line represents the mean value (F–I). Scale bars, 10 $\mu m$ (E), 30 $\mu m$ (C, D and I-H&E), 3 mm (I-GFP image), and 5 mm (F). See also Fig S2. Source data are available for this figure.

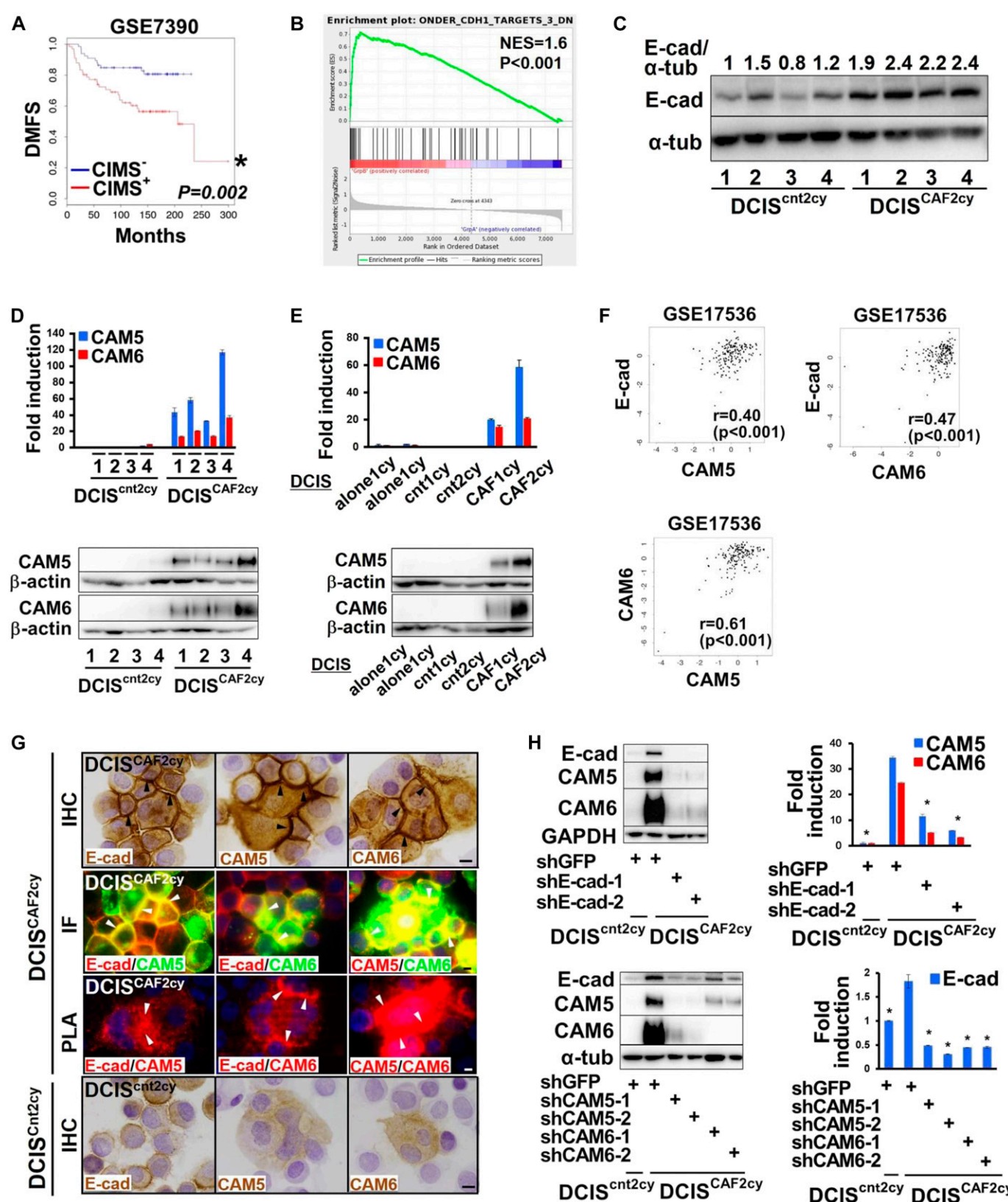

**Figure 3. The E^hi state mediated by E-cad, CAM5, and CAM6 expressions in DCIS^CAF2cy.**
**(A)** Kaplan–Meier survival analysis for distant metastasis-free survival (DMFS) using the CAF-induced metastasis signature (CIMS) in the human breast cancer patient cohort GSE7390. **(B)** Gene set enrichment analysis in DCIS^CAF2cy relative to DCIS^cnt2cy. DCIS^CAF2cy shows the enrichment of genes down-regulated in E-cad-shRNA–expressing breast

## Src activation mediates $E^{hi}$ and E/M states, invasion, and metastasis in DCIS$^{CAF2cy}$

As the $E^{hi}$ and E/M states have crucial roles in metastasis formation of DCIS$^{CAF2cy}$, signaling pathways regulating these states were explored using high-throughput screening. We found PP1 analog, an Src inhibitor, to be a promising candidate, based on achieving suppression of CAM6 mRNA expression exceeding 50% in DCIS$^{CAF2cy}$ (Fig S5A). Endogenous expression levels of phosphorylated Src (p-Src) were also markedly increased in DCIS$^{CAF2cy}$ as compared with DCIS$^{cnt2cy}$ (Figs 6A and S5B). We thus investigated the roles of Src signaling in DCIS$^{CAF2cy}$ using saracatinib, a different Src-family kinase inhibitor (Fig 6A), and two Src-shRNAs, both of which significantly inhibited Src expression (Figs 6B and S5C). Each of these treatments attenuated CAM5, CAM6, E-cad, and p-Src expressions significantly more than did DMSO or GFP-shRNA (Figs 6A and B and S5D and E), indicating Src activation to be required for maintenance of the $E^{hi}$ state on DCIS$^{CAF2cy}$.

As a previous report indicated that Src activation is induced by CAM6 in pancreatic cancer cells (Duxbury et al, 2004), we assumed a self-activated mechanism(s) in Src signaling in DCIS$^{CAF2cy}$. As speculated, inhibition of CAM5, CAM6, or E-cad expression by shRNA notably attenuated p-Src expression in these cells as compared with the effect of GFP-shRNA (Fig 6C). Our data demonstrate CAM5, CAM6, and E-cad expressions to be up-regulated via Src activation, which in turn activates Src signaling in a self-stimulating autocrine fashion in DCIS$^{CAF2cy}$. Importantly, greater CAM5, CAM6, E-cad, and p-Src protein expressions were also semi-stably maintained in these cells during their in vitro propagation up to 15 population doublings in pure culture (Fig 6D), thereby revealing the establishment of self-stimulating autocrine signaling to retain the $E^{hi}$ state in DCIS$^{CAF2cy}$ even in the absence of ongoing interactions with CAFs.

Because E-cad is required for cell–cell adhesion and collective cell migration to activate Src (McLachlan et al, 2007; Veracini et al, 2015), we reasoned that E-cad contributes to activation of Src signaling on DCIS$^{CAF2cy}$. E-cad was indeed co-localized and associated with Src on these cells, as gauged by immunostaining and in situ PLA, respectively (Figs 6E and S5F). In contrast, CAM5 and CAM6, glycosylphosphatidylinositol anchored proteins lacking the intracellular domain, failed to co-localize with Src (Fig S5G), indicating Src signaling to presumably be mediated by E-cad on DCIS$^{CAF2cy}$.

Given that ZEB1 expression mediating the E/M state is up-regulated in DCIS$^{CAF2cy}$, we examined whether Src activation is also required for ZEB1 expression in these cells. Treatment with Src-shRNA or saracatinib attenuated ZEB1 expression significantly more than did GFP-shRNA or DMSO (Figs 6F and S5H), indicating ZEB1 expression to be mediated by Src activation in DCIS$^{CAF2cy}$.

We next investigated the effects of Src activation on biological properties of DCIS$^{CAF2cy}$. Inhibition of Src activation by saracatinib or PP1 attenuated proliferation, invasion, cell–cell adhesion, and cell–cell aggregation in these cells more significantly than did DMSO (Fig 6G–J). Furthermore, lung-colonizing ability was significantly inhibited in DCIS$^{CAF2cy}$ expressing Src-shRNA as compared with control GFP-shRNA (Fig 6K). These findings indicate that Src activation is required for various tumor- and metastasis-promoting properties presumably mediated by the $E^{hi}$ and E/M states in DCIS$^{CAF2cy}$.

Because Src activation is required for tumor metastasis in DCIS$^{CAF2cy}$, we investigated whether Src activation induces metastasis by introducing a constitutively active Src mutant (Kano et al, 2008) into parental DCIS cells (Fig S5I). Lung-colonizing ability was moderately elevated in these cells (Fig S5J), but Src activation by itself was not sufficient to fully induce the metastatic property of DCIS$^{CAF2cy}$ in parental DCIS cells.

## Stromal cell-derived factor 1 (SDF-1) and transforming growth factor-$\beta$ (TGF-$\beta$) mediate $E^{hi}$ and E/M tumor cell cluster formation, invasion, and metastasis via Src activation

CAFs have previously been demonstrated to produce high levels of SDF-1 and TGF-$\beta$, contributing to the growth and progression of nearby carcinoma cells by acting in a paracrine fashion (Orimo et al, 2005; Kojima et al, 2010; Zhang et al, 2013; Scherz-Shouval et al, 2014). We thus reasoned that these stromal cytokines might mediate CAF-induced metastasis. Our previous work showed that introduction of SDF-1-shRNA into CAFs attenuates their SDF-1 and TGF-$\beta$ mRNA expressions because of disruption of the cross-communicating SDF-1 and TGF-$\beta$ autocrine signaling loop on these fibroblasts (Kojima et al, 2010). Inhibiting TGF-$\beta$ autocrine signaling by the TGF-$\beta$ receptor II extracellular (T$\beta$RII ecto) domain construct (Thomas & Massague, 2005) also significantly suppressed the SDF-1 and TGF-$\beta$ mRNA expressions in CAFs (Fig 7A). We thus injected CAFs expressing the SDF-1-shRNAs or the T$\beta$RII ectodomain construct with DCIS cells subcutaneously into mice (Fig 7B). Notably, inhibition of SDF-1 and TGF-$\beta$ expressions by each of these constructs in CAFs significantly attenuated the lung metastatic index (Fig 7C).

We next investigated whether CAF-produced SDF-1 and TGF-$\beta$ are required for induction of epithelial–mesenchymal plasticity and Src activation in DCIS cells. We thus extracted the injected DCIS cells, designated DCIS$^{cnt2cy}$ or DCIS$^{CAF2cy}$, from the developing subcutaneous tumor xenografts admixed with control fibroblasts (expressing GFP) or CAFs (expressing GFP or the T$\beta$RII ecto), respectively, in mice (Fig 7B). The inhibition of SDF-1 and TGF-$\beta$ expressions in CAFs expressing the T$\beta$RII ecto significantly attenuated the $E^{hi}$ and E/M states, as exemplified by CAM5, CAM6, E-cad, and ZEB1, and p-Src expressions in DCIS$^{CAF2cy}$, as compared with the effect

---

cancer cells (Onder et al, 2008). **(C)** Immunoblotting of the described cells extracted from four different tumors (1–4) using the indicated antibody. The signal intensity ratios of E-cad relative to $\alpha$-tubulin are indicated. **(D)** Real-time PCR (top) and immunoblotting (bottom) of DCIS$^{cnt2cy}$ and DCIS$^{CAF2cy}$ extracted from four different tumor xenografts measuring the indicated gene expressions. **(E)** Real-time PCR (top) and immunoblotting (bottom) of the described cells measuring the indicated gene expressions. **(F)** Positive linear correlations between E-cad, CAM5, and CAM6 mRNA expressions in the human breast cancer cohort GSE17536. **(G)** Immunostaining and in situ PLA in the indicated cells using the depicted antibodies. Positive staining (arrowhead) is shown on adherence junctions between DCIS$^{CAF2cy}$. Scale bars, 10 $\mu$m. **(H)** Immunoblotting (left) and real-time PCR (right) of the indicated cells. Data information: Asterisk indicates a significant difference relative to the CIMS$^-$ group (A) and GFP-shRNA–expressing DCIS$^{CAF2cy}$ (H). t test (H) and Cox proportional hazards regression test (A). Error bars, SE. See also Fig S3 and Table S1. IHC, immunohistochemistry; IF, immunofluorescence; PLA, in situ PLA. Source data are available for this figure.

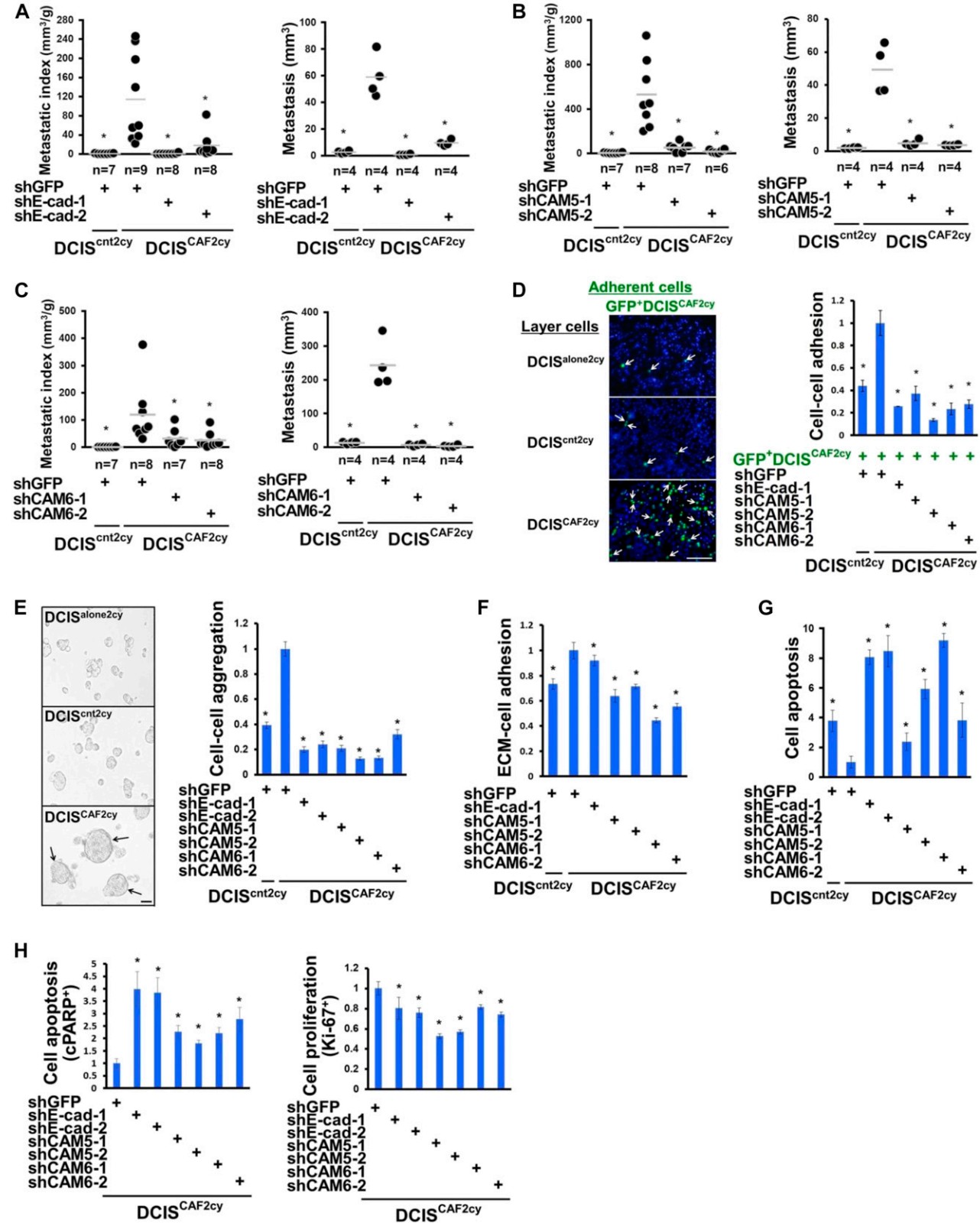

**Figure 4.  The E^hi state required for invasive and metastatic abilities in DCIS^CAF2cy.**
**(A–C)** Lung metastatic indices (left) evaluated at 60 d after subcutaneous injection of the indicated cells into mice. Lung metastatic volume (right) was evaluated at 30 d after intravenous injection of the indicated cells into mice. **(D)** Appearance of GFP⁺DCIS^CAF2cy (arrows) attached to the top layer of the indicated cells (left). The relative numbers of GFP⁺DCIS^CAF2cy attached to the indicated cells (n = 4) are shown (right). **(E)** Appearance of larger cell aggregates (arrows) formed by DCIS^CAF2cy compared to

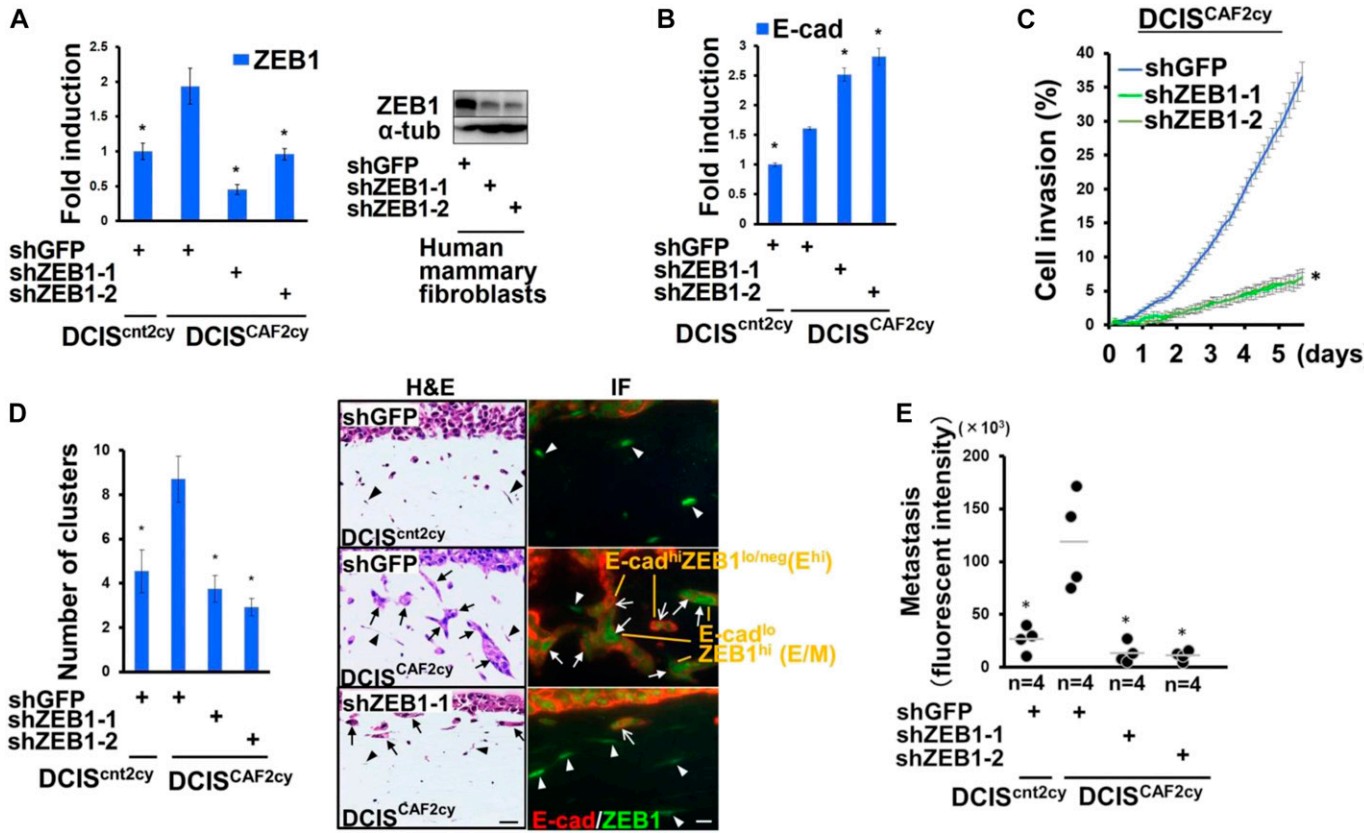

**Figure 5. The E/M state mediates invasive and metastatic abilities in DCIS^CAF2cy.**
**(A)** Real-time PCR of the indicated cells measuring ZEB1 expression (left). Immunoblotting of the described cells using anti-ZEB1 and anti–$\alpha$-tub antibodies (right). **(B)** Real-time PCR of the indicated cells measuring E-cad expression. **(C)** Cell invasion evaluated by scratch wound assay (n = 8) using the indicated cells. **(D)** Cell invasion evaluated by organotypic invasion assay using the indicated cells (n = 3). H&E staining (right-left) of the organotypic gel containing the indicated cancer cells (arrows) and mammary fibroblasts (arrowheads), and its immunofluorescence (right-right) using anti–E-cad and anti-ZEB1 antibodies. The E^hi (simple arrow) and E/M (triangular arrow) cancer cells, and nuclear ZEB1+ stromal cells (arrowheads) are shown (right-right). Scale bars, 30 $\mu$m (H&E), 10 $\mu$m (IF). **(E)** Lung metastases evaluated by fluorescent intensity at 30 d after intravenous injection of the indicated tdTomato+ cells into mice. Data information: Asterisk indicates a significant difference relative to GFP-shRNA–expressing DCIS^CAF2cy (A–E). *t* test (A–D) and Mann–Whitney *U* test (E). The horizontal line represents the mean value (E). Error bars, SE. IF, immunofluorescence.

Source data are available for this figure.

---

of control GFP (Figs 7D and S6A). Taken together, these data indicate that stromal SDF-1 and TGF-$\beta$ are required for induction of E^hi and E/M states, Src activation, and metastatic ability in DCIS^CAF2cy.

We next examined whether stromal SDF-1 and TGF-$\beta$ act through their cognate receptors CXCR4 and T$\beta$RII, respectively, expressed on nearby DCIS cells. DCIS cells expressing GFP-, CXCR4-, or T$\beta$RII-shRNA (Kojima et al, 2010) were then injected with CAFs subcutaneously into recipient mice (Fig 7E). Inhibition of CXCR4 or T$\beta$RII expression by shRNA in tumor cells significantly attenuated the lung metastatic index resulting from the actions of CAFs, as compared with the effect of GFP-shRNA (Fig 7F). CAM5, CAM6, E-cad, and ZEB1 expressions mediating the E^hi and E/M states were also significantly attenuated in DCIS^CAF1cy extracted from DCIS tumors expressing CXCR4- or T$\beta$RII-

shRNA admixed with CAFs (Figs 7E, G, H, and S6B). Collectively, these data indicate stromal SDF-1 and TGF-$\beta$ to be required for induction of the E^hi and E/M states, Src activation, and metastatic traits via their cognate receptors present on DCIS cells.

### Stromal SDF-1 and TGF-$\beta$ give rise to E^hi and E/M states and collective invasion of breast cancer cell clusters through Src

We next investigated whether stromal SDF-1 and TGF-$\beta$ initiate the E^hi and E/M states in human breast cancer cells via Src activation. DCIS cells were thus treated with recombinant SDF-1 and/or TGF-$\beta$1. Treatment with both SDF-1 and TGF-$\beta$1 increased CAM6, E-cad, and ZEB1 expressions (Figs 7I and S6C) and the resulting E^hi and E/M cell

---

DCIS^alone2cy and DCIS^cnt2cy on low attachment culture dishes (left). The relative volume of cell aggregates is shown in the indicated cells (n = 5) (right). **(F)** The ECM (collagen)–cell adhesion measured in the indicated cells (n = 4–9). **(G)** Cell apoptosis measured in the indicated cells (n = 5) on low attachment culture dishes. **(H)** The relative apoptotic (left) and proliferating (right) tumor cell proportions in experimental lung metastases generated by DCIS^CAF2cy expressing the indicated shRNA (n = 4) by immunostaining using anti-cPARP and anti–Ki-67 antibodies, respectively. Data information: Asterisk indicates a significant difference relative to GFP-shRNA–expressing DCIS^CAF2cy (A–H). *t* test (D–H) and Mann–Whitney *U* test (A–C). Error bars, SE. The horizontal line represents the mean value (A–C). Scale bars, 30 $\mu$m (E) and 300 $\mu$m (D). See also Fig S4.

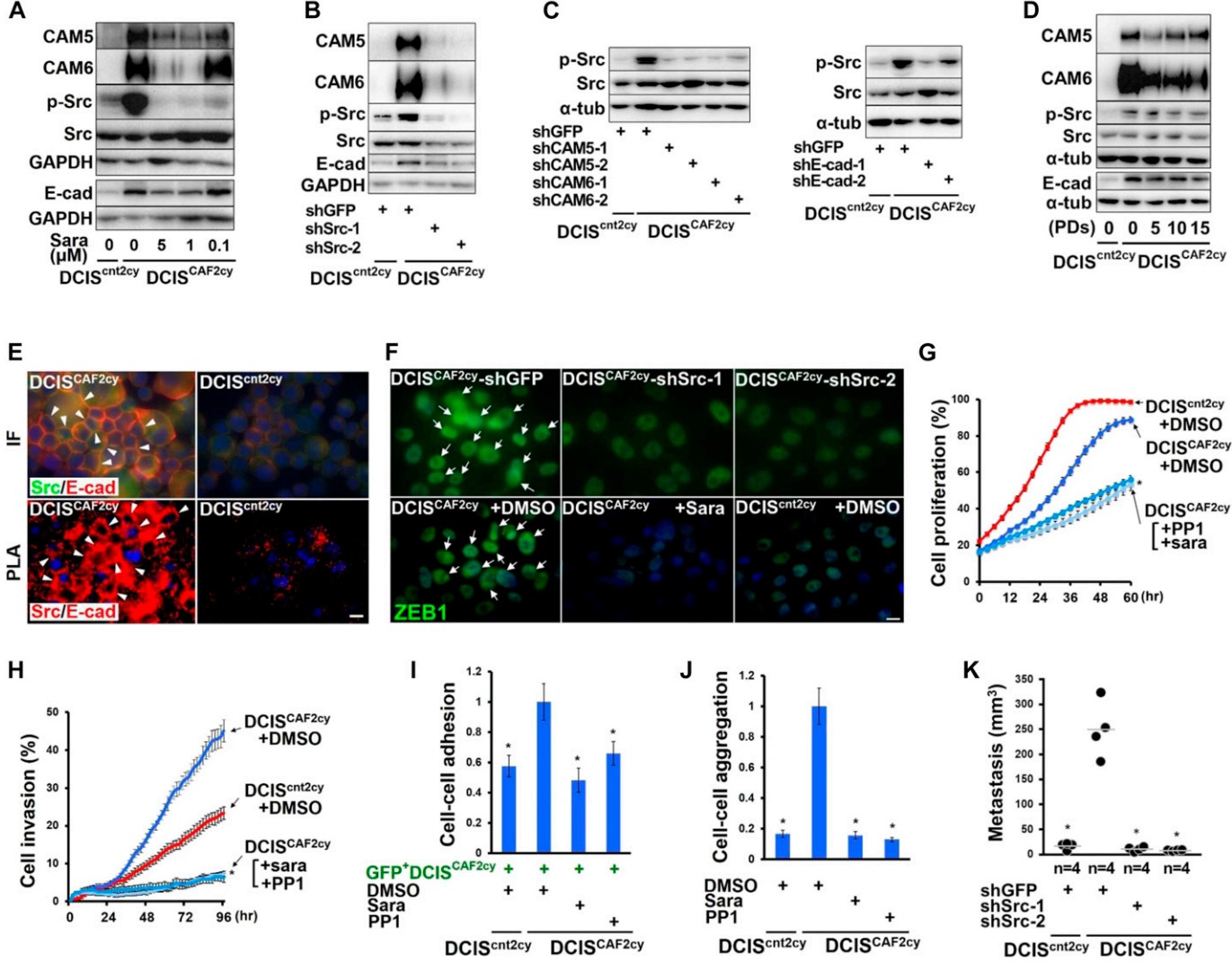

**Figure 6. Src activation mediates the E[hi] and E/M states and metastatic ability in DCIS[CAF2cy].**
**(A)** Immunoblotting of the described cells treated with different concentrations of saracatinib (sara) dissolved in DMSO for 24 h using the indicated antibodies. **(B, C)** Immunoblotting of the indicated cells using the described antibodies. **(D)** Immunoblotting of the indicated cells passaged up to 15 population doublings (PDs) using the described antibodies. **(E)** Immunostaining (IF) and in situ PLA (PLA) in the indicated cells using anti-Src and anti–E-cad antibodies. Positive staining (arrowhead) is shown on adherence junctions between DCIS[CAF2cy]. **(F)** Immunofluorescence of the indicated cells using an anti-ZEB1 antibody. Nuclear ZEB1+ cells (arrows) are also shown. **(G–J)** (G) Cell proliferation, (H) scratch wound cell invasion, (I) cell–cell adhesion, and (J) cell–cell aggregation in the indicated cells treated with DMSO, saracatinib (1 μM) or PP1 (10 μM) (n = 4–8). **(K)** Lung metastasis evaluated at 30 d after intravenous injection of the indicated cells into mice. The horizontal line represents the mean value. Data information: Asterisk indicates a significant difference relative to DMSO-treated DCIS[CAF2cy] (G–J) and the GFP-shRNA–expressing DCIS[CAF2cy] (K). t test (G–J) and Mann–Whitney U test (K). Error bars, SE. Scale bars, 10 μm (E, F). See also Fig S5.
Source data are available for this figure.

proportions (Fig 7J) in GFP-shRNA–expressing DCIS cells significantly more than did PBS. Inhibition of Src expression by shRNA significantly attenuated these observed inductions in DCIS cells treated with SDF-1 and TGF-β1 (Fig 7I and J), indicating Src to be required for induction of the E[hi] and E/M states by stromal SDF-1 and TGF-β. The E[hi] and E/M cell proportions were also larger in DCIS cells expressing active Src mutant cDNA as compared with those expressing the control vector (Fig 7K). Furthermore, SDF-1 and TGF-β1 treatment stimulated collective invasion of DCIS cell clusters with the E[hi] and E/M states significantly more than did PBS (Fig 7L). In another human breast cancer cell line, MCF-7-ras cells, CAM6, E-cad, and ZEB1 mRNA expressions were also consistently induced

by SDF-1 and TGF-β1 treatment (Fig S6D). These findings, along with earlier observations, indicate that CAF-produced SDF-1 and TGF-β induce and maintain the E[hi] and E/M states as well as collective invasive and metastatic abilities in breast carcinoma cells via Src activation during tumor progression.

### CAF-induced CTC clusters, tumor emboli, and lung colonization during metastasis

As CAFs promoted cell–cell adhesion and aggregation of cultured tumor cells (Fig 4D and E), we reasoned that these fibroblasts induce formation of tumor cell clusters in vivo. To investigate this

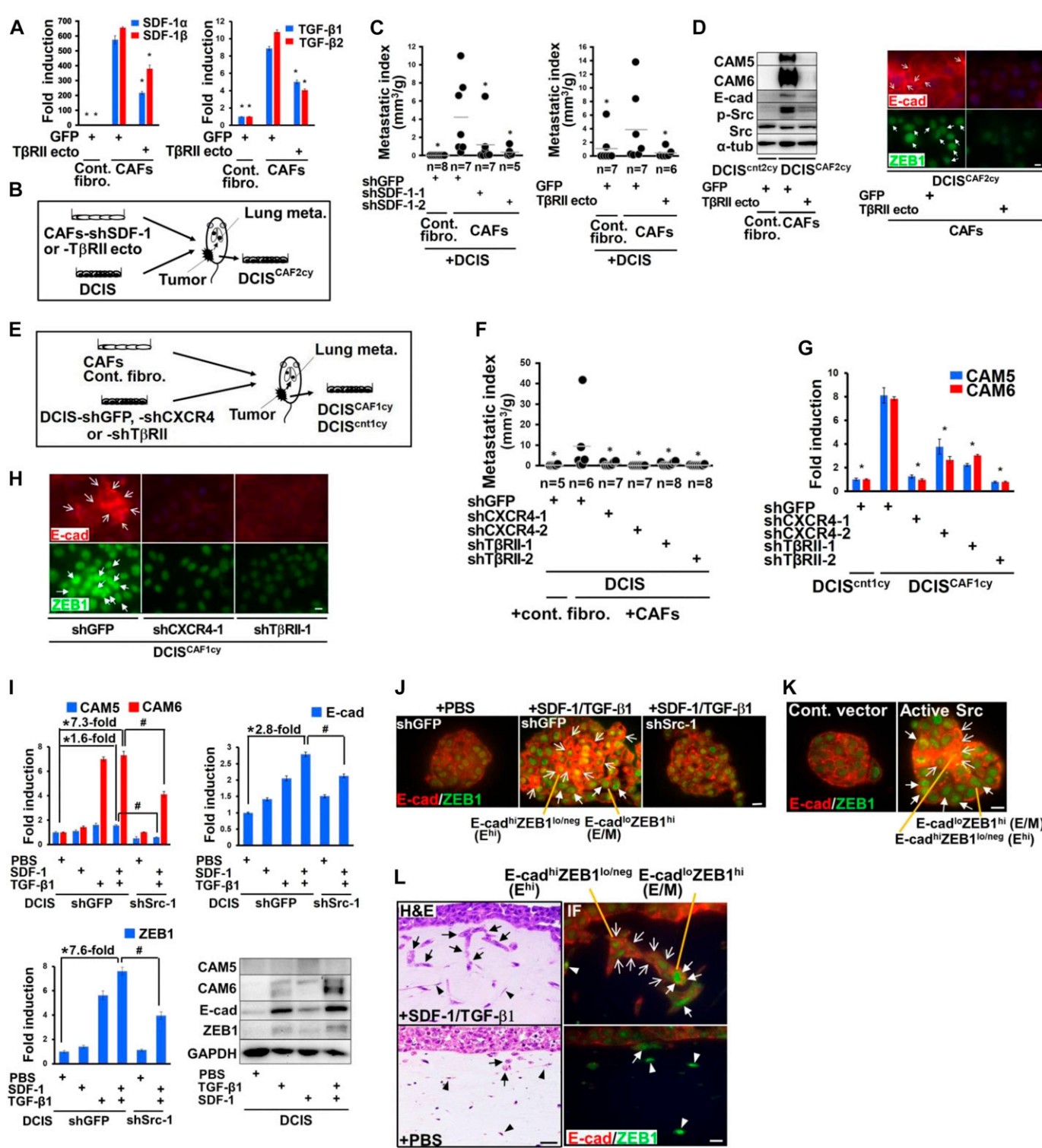

**Figure 7. Stromal SDF-1 and TGF-β mediate the formation of invasive and metastatic breast tumor clusters with E^hi and E/M states via Src activation.**
**(A)** Real-time PCR of control fibroblasts and CAFs expressing GFP or TβRII ecto using the indicated primers. **(B)** Schematic illustration of a subcutaneous co-implantation tumor xenograft model (for Fig 7C and D). *See text in detail.* **(C)** The lung metastatic indices are measured at 60 d after subcutaneous injection of the indicated cells. **(D)** Immunoblotting of DCIS^cnt2cy and DCIS^CAF2cy using the indicated antibodies (left). The DCIS^cnt2cy and DCIS^CAF2cy were extracted from tumors generated by DCIS cells admixed with control fibroblasts (expressing GFP) and CAFs (expressing GFP or TβRII ecto), respectively. Immunostaining of the indicated cells using anti–E-cad and anti-ZEB1 antibodies (right). E-cad^+ cancer cells (simple arrows) and nuclear ZEB1^+ cancer cells (triangular arrows) are shown. **(E)** Schematic illustration of a subcutaneous co-implantation tumor xenograft model (for Fig 7F–H). *See text for details.* **(F)** The lung metastatic indices evaluated at 60 d after subcutaneous injection of the described cells into mice. **(G)** Real-time PCR of DCIS^cnt1cy or DCIS^CAF1cy measuring the indicated gene expressions. The DCIS^cnt1cy and DCIS^CAF1cy were extracted from 30-d-old tumor

possibility, peripheral blood was taken from mice, which had been injected subcutaneously with tdTomato+ DCIS cells and different fibroblasts followed by immunostaining and colony-formation assay. E-cad+, CAM5+, CAM6+, and ZEB1+ CTC clusters were detected in circulating blood from mice bearing tumors admixed with CAFs (Fig 8A). CTCs derived from tumors admixed with CAFs, as compared with no fibroblasts and control fibroblasts, also notably formed larger numbers of tdTomato+ colonies in culture, when seeded onto petri dishes (Fig 8B). Moreover, DCIS^CAF2cy, when injected subcutaneously into mice, frequently generated E-cad+ and Ki-67+ tumor emboli within α-SMA+ blood vessels in the lungs (Figs 8C and S7A), further supporting the CAF-induced tumor cell cluster formation in vivo. Collectively, these findings demonstrate that CAFs stimulate the formation of tumor cell clusters presumably with E^hi and E/M states in the peripheral bloodstream, thereby seeding metastasis.

We next examined whether the CAF-primed E^hi state in subcutaneous tumors and CTC clusters (Figs 1A, 2D, and 8A) is retained during metastatic colonization of distant organs. Thus, lung sections prepared from mice injected subcutaneously with DCIS^CAF2cy or DCIS^cnt2cy were stained with different antibodies. Of note, E-cad+, CAM5+, CAM6+, and p-Src+ lung metastatic nodules were significantly larger and more numerous with DCIS^CAF2cy than with DCIS^cnt2cy (Fig 8D), indicating maintenance of the CAF-primed E^hi state during metastatic dissemination and colonization. Consistently, the E^hi state in primary tumors was also retained in corresponding lung metastases in mice subcutaneously injected with DCIS cells and CAFs (Fig S7B).

Given that ZEB1− pulmonary tumor emboli occur in mice in response to DCIS^CAF2cy (Fig 8C), we reasoned that the mesenchymal trait in E/M tumor cells is down-regulated through MET during metastatic colonization. As anticipated, increased ZEB1 staining in primary tumors due to DCIS^CAF2cy (Fig 2D), was dramatically attenuated in the accompanying lung metastases (Fig 8D). Consistently, ZEB1 staining was also down-regulated in lung metastases produced by DCIS cells admixed with CAFs subcutaneously injected into mice (Fig S7B), indicating attenuation of mesenchymal trait during metastatic colonization.

Moreover, in the human breast cancer metastasis cohort with lung, liver, and bone metastases, the positive linear correlation between CAM6 and CAM5 mRNA expressions with concomitant E-cad expression was significant, whereas ZEB1 expression correlated negatively with E-cad expression (Fig 8E). These data indicate that the CAF-induced E^hi state is retained in tumor cell clusters circulating, disseminating, and colonizing the lungs, whereas the mesenchymal trait in E/M tumor cells is down-regulated presumably through MET during metastatic colonization (Fig 8F).

### CAF-induced E^hi and E/M states are associated with the Her2+ER−PR− tumor status and poor outcomes in breast cancer patients

We next sought to determine whether E^hi and E/M tumor cell clusters are detectable in breast cancer patients. The human breast cancer tissues were thus stained with anti–E-cad and anti-ZEB1 antibodies. Microvascular tumor emboli containing E^hi and E/M tumor cells were observed in the breast parenchyma (Fig 9A), indicating E^hi and E/M tumor cell clusters to indeed be present in breast cancer patients.

The clinical relevance of CAF-induced E^hi and E/M states was further examined in breast cancer subtypes using immunohistochemistry on 257 human breast cancers. Positive staining for CAM6/CAM5/E-cad and E-cad/ZEB1 was detected in Her2+ER−PR− (Her2-positive, estrogen receptor–negative, and progesterone receptor-negative) breast cancers more frequently than in other tumors (Fig 9B and Tables S2, S3, S4, S5). In addition, E^hi and E/M tumor cell populations were detected in Her2+ER−PR− human breast tumor sections (Fig 9C). These data indicate the CAF-induced E^hi and E/M states to be significantly associated with Her2+ER−PR− human breast carcinomas.

The relevance of E^hi and E/M states to disease prognosis was next examined using a publicly available database. The high CAM6/CAM5/E-cad and E-cad/ZEB1 mRNA expressions were significantly associated with poorer relapse-free survival in the Her2+ER−PR−, but not in the Her2−ER+PR+ and Her2−ER−PR−, breast cancer patient cohorts (Figs 9D and S8A). Collectively, these observations indicate that CAF-induced E^hi and E/M states are associated with poor survival in Her2+ER−PR− breast cancer patients.

## Discussion

### Stromal SDF-1 and TGF-β drive E^hi and E/M tumor cell cluster formation via Src activation to seed metastasis

Although complete EMT plays central roles in creating solitary mesenchymal tumor cells that disseminate and seed metastasis (Thiery et al, 2009), recent emerging evidence supports the observation that tumor cell clusters seed metastasis significantly

---

xenografts generated by DCIS cells expressing the indicated shRNA admixed with control fibroblasts and CAFs, respectively. **(H)** Immunostaining of the described cells using the indicated antibodies. E-cad+ cancer cells (simple arrows) and nuclear ZEB1+ cancer cells (triangular arrows) are shown. **(I)** Real-time PCR of DCIS cells expressing the indicated shRNA, treated with PBS, SDF-1 (100 ng/ml), and/or TGF-β1 (10 ng/ml) for 24 h, to measure the described gene expressions. Immunoblotting of DCIS cells treated with PBS, SDF-1, and/or TGF-β1 for 48 h using the indicated antibodies (right-bottom). **(J)** Immunofluorescence of DCIS organoids expressing the indicated shRNAs treated with PBS or both SDF-1 (100 ng/ml) and TGF-β1 (10 ng/ml) using anti–E-cad and anti-ZEB1 antibodies. The E^hi (simple arrow) and E/M (triangular arrow) cancer cells are also shown. **(K)** Immunofluorescence of DCIS organoids expressing the control empty vector (Cont. vector) or the constitutively active Src mutant (Active Src) using the indicated antibodies. The E^hi (simple arrow) and E/M (triangular arrow) cancer cells are shown. **(L)** Organotypic invasion assay using DCIS cells treated with PBS or both SDF-1 (100 ng/ml) and TGF-β1 (10 ng/ml). Invading tumor cell clusters (arrows) and mammary fibroblasts (arrowheads) embedded in the gel with H&E staining (left) and immunofluorescence (IF) using anti–E-cad and anti-ZEB1 antibodies (right) are shown. The E^hi (simple arrow) and E/M (triangular arrow) cancer cells, and nuclear ZEB1+ stromal cells (arrowheads) are shown (right). Data information: Asterisk indicates a significant difference relative to GFP-expressing CAFs (A and C-right), GFP-shRNA–expressing CAFs (C-left), and GFP-shRNA–expressing DCIS cells (F) and DCIS^CAF1cy (G). Asterisk and # symbol also indicate a significant difference between the indicated lines (I). t test (A, G, I) and Mann–Whitney U test (C, F). Error bars, SE. The horizontal line represents the mean value (C, F). Scale bars, 30 μm (L-H&E) and 10 μm (others). See also Fig S6.
Source data are available for this figure.

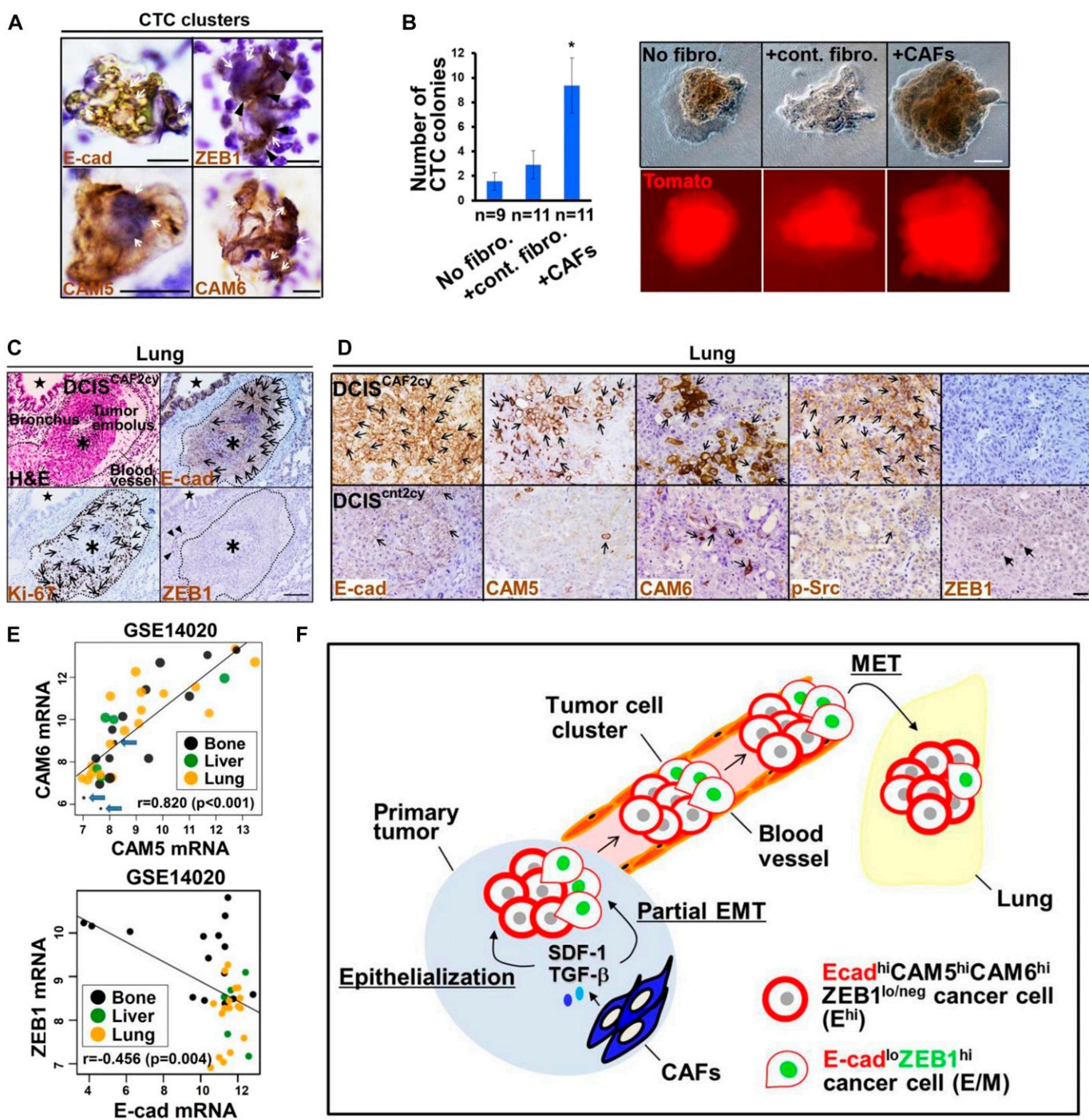

**Figure 8. CAF-induced CTC clusters, tumor emboli, and metastatic colonization.**
**(A)** Immunostaining of cytospin-concentrated smears prepared from peripheral blood of mice bearing 30-d-old DCIS tumors admixed with CAFs using the indicated antibodies. Positive staining (brown) for E-cad, CAM5, CAM6, and ZEB1 and hematoxylin nuclear staining (blue) indicated by arrows are shown in cancer cells of CTC clusters. Nuclear ZEB1 staining (arrowhead) is also depicted in tumor cells. A number of leukocytes around the CTC clusters are stained with hematoxylin. **(B)** Number of CTC colonies evaluated by colony-forming assay (left). Peripheral blood was collected from mice injected subcutaneously with tdTomato-labelled, blasticidin-resistant DCIS cells admixed with no fibroblasts (No fibro.), control fibroblasts (+cont. fibro.), or CAFs (+CAFs) before culture in the presence of blasticidin on a petri dish for 21 d. Visualization of blasticidin-resistant CTC colonies (right-upper) and their tdTomato positivity under fluorescent microscopy (right-lower). Asterisk indicates a significant difference relative to No fibro. and +cont. fibro. groups. Error bars, SE. **(C)** H&E staining and immunohistochemistry of lung sections prepared at 60 d after subcutaneous injection of DCIS^CAF2cy into mice using the indicated antibodies. E-cad+ and Ki-67+ carcinoma cells (arrows) are indicated in tumor emboli (asterisk in broken circle). E-cad+ epithelial cells are also shown in a bronchus (star), as well as nuclear ZEB1+ mesenchymal cells (arrowheads). **(D)** Immunostaining of lung sections prepared at 60 d after subcutaneous injection of DCIS^CAF2cy or DCIS^cnt2cy into mice using the indicated antibodies. E-cad+, CAM5+, CAM6+, and p-Src+ cancer cells (simple arrows) as well as nuclear ZEB1+ cancer cells (triangular arrows) are shown. **(E)** Linear correlations between the indicated genes expressed in metastatic sites, including the bone, liver, and lung in the breast cancer patient cohort GSE14020. E-cad expression is represented by the sizes of circles including the three patients with the lowest E-cad expressions

more than do the single tumor cells (Aceto et al, 2014; Maddipati & Stanger, 2015; Cheung et al, 2016). The formations of invading tumor cell clusters are also attributable to partial EMT (Westcott et al, 2015; Campbell & Casanova, 2016; Grigore et al, 2016; Aiello et al, 2018; Li et al, 2019). CAF abundance in the tumor microenvironment promotes tumor invasion and metastasis (Calon et al, 2012; Zhang et al, 2013). However, the relationships between tumor cell clusters, partial EMT, and CAF-primed metastasis remain unclear.

We herein describe a previously unrecognized role of stroma-induced epithelial–mesenchymal plasticity in the invading tumor cell cluster formation; a highly epithelial nature with partial EMT is induced by CAFs to form tumor cell clusters such that they markedly promote collective cell invasion and metastatic seeding in vivo. The CAF-primed multicellular cluster is also composed of two distinct tumor cells, one in the E[hi] state and another in the E/M state in tumor xenografts admixed with CAFs (Fig 8F). Moreover, the E[hi] and E/M tumor cell clusters are seen in the primary human breast carcinoma (Fig 9A).

We have demonstrated herein that CAF-primed E[hi] tumor cells, mediated by elevated E-cad, CAM5, and CAM6 expressions, boost cell–cell adhesion, aggregation, and antiapoptosis, allowing the formation of multicellular clusters and promoting metastatic seeding. The CAF-primed E[hi] state is also stably maintained on breast cancer cells via Src activation in a self-stimulating autocrine fashion during tumor progression (Fig 6D). Analysis of DNA methylation by pyrosequencing reveals the slightly hypomethylated promoter region present in the CAM6 gene, but not in either the CAM5 or the E-cad gene, in DCIS[CAF2cy] as compared with DCIS[cnt2cy] (Fig S9A), indicating minimal involvement of DNA methylation. Other epigenetic alterations regulating histone modifications may be attributable to maintenance of the E[hi] state in DCIS[CAF2cy].

We have also shown induction of ZEB1 expression to generate the E/M state presumably by partially suppressing E-cad expression in CAF-primed tumor cells (Fig 5B). The E/M state plays crucial roles, as it provides leader cells that retain associations with the follower E[hi] tumor cell clusters leading to their collective cell invasion. The ZEB1 expression might also take advantage of tumor progression by various means related to metastasis initiation, stemness, cellular survival, and metabolic plasticity (Puisieux et al, 2014; Krebs et al, 2017). Moreover, the disseminating tumor cell clusters boost metastatic colonization of the lungs via MET, as exemplified by attenuated ZEB1 expression. These observations indicate that E[hi] and E/M tumor cell clusters induced by CAFs collectively invade and seed metastasis.

Our data also show that the E[hi] and E/M states are induced and maintained by SDF-1 and TGF-$\beta$ released from CAFs via Src activation required for collective tumor cell invasion and metastasis. Treatment with both SDF-1 and TGF-$\beta$1 strongly up-regulates CAM6, E-cad, and ZEB1 expressions (Fig 7I) and enhances the E[hi] and E/M tumor cell proportions (Fig 7J) in DCIS cells through Src, leading to increased collective invasion (Fig 7L). These data are supported by previous reports indicating that SDF-1 plus TGF-$\beta$ treatment induces

epithelial–mesenchymal plasticity (Li et al, 2014; Yu et al, 2014) and Src activation (Cabioglu et al, 2005; Chinni et al, 2008; Wang et al, 2009) in cultured tumor epithelial cells. Although TGF-$\beta$ is a well-known inducer of complete EMT, strongly inhibiting E-cad expression (Thiery et al, 2009), treatment with this cytokine reportedly up-regulates E-cad expression in human colorectal cancer cell organoids and epithelial Langerhans cells in culture (Riedl et al, 2000; Calon et al, 2015). SDF-1 treatment also up-regulates E-cad expression and stimulates its relocation to the cell–cell membrane in colon cancer cells during collective migration (Hwang et al, 2012). These observations are consistent with our findings in different breast cancer cells, indicating the cell context-dependent functions of these cytokines.

In CAF-admixed breast tumor xenografts, we observed the induction of the E/M state as evidenced by ZEB1 expression in tumor cells adjacent to CAFs at the tumor–stroma interface (Fig 1B). These findings are in accordance with those of a previous report describing the partial EMT program in human head and neck cancer cells as being closely associated with CAFs (Puram et al, 2017). CAF-produced SDF-1 and TGF-$\beta$ are likely to be attributable to this induction, via Src, in these carcinoma cells. Moreover, stromal SDF-1 and TGF-$\beta$ also contribute to an enhanced E[hi] state with Src activation in cancer cells. Forced Src activation does, in fact, increase proportions of both E[hi] and E/M tumor cells (Fig 7K). Induction of either the E[hi] or the E/M state may, therefore, depend on differing responsiveness to Src activation of heterogeneous cancer cell populations exposed to SDF-1 and TGF-$\beta$ cytokines. We also speculate that E[hi] and E/M are indeed interconvertible cell states in a tumor cell cluster during tumor progression. Future investigation of these cell populations at the single-cell level may allow us to determine their precise roles in CAF-induced epithelial–mesenchymal plasticity.

## CAF-primed E[hi] and E/M states are associated with Her2⁺ER⁻PR⁻ status and poor outcomes in breast cancer patients

The clinical significance of E[hi] and E/M states, in human breast carcinomas, is poorly understood. Our analyses revealed CAF-induced E[hi] and E/M states, as exemplified by CAM6/CAM5/E-cad and E-cad/ZEB1 expressions in DCIS cells, respectively, to be associated with both the Her2⁺ER⁻PR⁻ status and poor outcomes of breast cancer patients (Figs 9B and D, and S8A). These findings suggest that there might be an interaction between CAFs and Her2⁺ER⁻PR⁻ breast carcinoma cells, enabling particular signaling pathways to cross-talk with one another, thereby promoting malignant tumor progression. Although the parental DCIS cells used in this study are known to only minimally express Her2 (Chung et al, 2016), we speculate that CAFs up-regulate Her2 expression in DCIS cells and/or increase the Her2-positive tumor cell subpopulation in tumors during tumor progression, facilitating their interaction with breast cancer cells. These possibilities need to be investigated in a future study.

Previous reports have described larger numbers of SDF-1–producing myofibroblasts to comprise tumor-associated stroma in

---

indicated by arrows (upper). **(F)** Schematic representation of CAF-induced invasive and metastatic tumor cell clusters composed of E[hi] and E/M tumor cells during the invasion-metastasis cascade. *See text for details.* Data information: Wilcoxon rank sum test (B) and two-sample correlation test (E). Scale bars, 10 $\mu$m (A), 1 mm (B), 100 $\mu$m (C), and 30 $\mu$m (D). See also Fig S7.

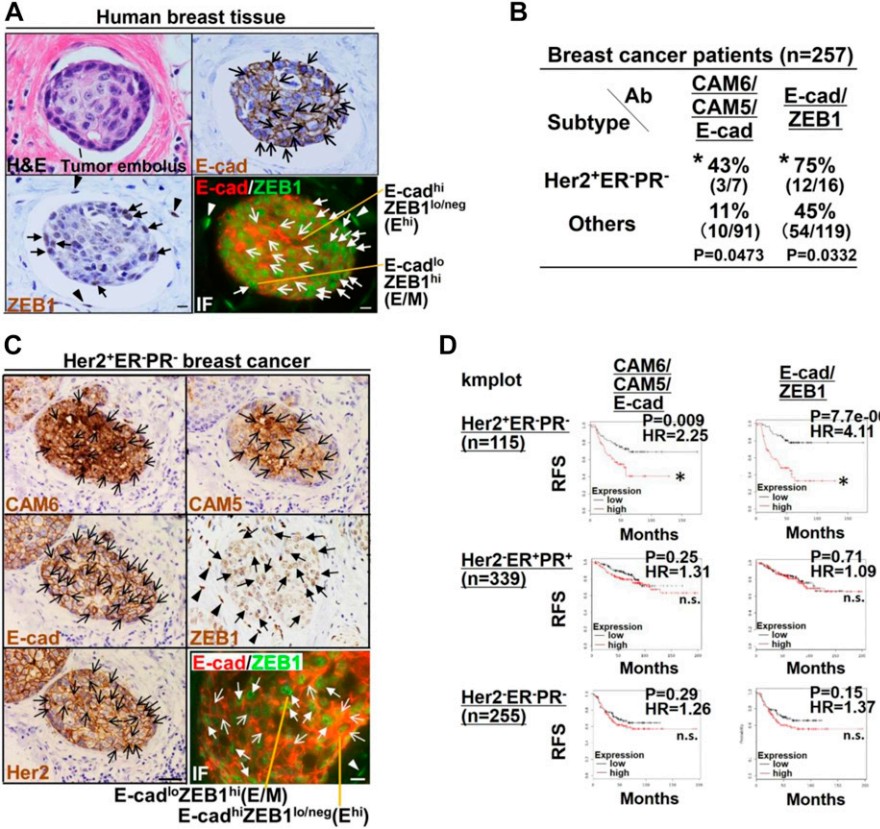

**Figure 9. The E^hi and E/M states in DCIS^CAF2cy are associated with poor outcomes for Her2^+ER^−PR^− breast cancer patients.**

**(A)** H&E (upper left) and immunostaining of tumor embolus present in a microvessel of human breast cancer tissue using the indicated antibodies. E-cad^+ cancer cells (simple arrows), nuclear ZEB1^+ cancer cells (triangular arrows), and nuclear ZEB1^+ stromal cells (arrowheads) are shown. The E^hi (simple arrow) and E/M (triangular arrow) cancer cells, and nuclear ZEB1^+ stromal cells (arrowheads) are also shown (lower right; IF, immunofluorescence). **(B)** Proportions (%) of breast cancer patients whose tumors stained positive for the indicated antibodies (Ab). Tumor sections that had been prepared from 257 breast cancer patients were immunohistochemically analyzed. The number of patients whose tumors stained positive for the indicated antibodies, relative to those stained both positive and negative, is shown in brackets. **(C)** Immunostaining of the Her2^+ER^−PR^− human breast carcinoma using the indicated antibodies. CAM6^+, CAM5^+, E-cad^+ and Her2^+ cancer cells (simple arrows), nuclear ZEB1^+ cancer cells (triangular arrows), and nuclear ZEB1^+ stromal cells (arrowheads) are shown. The E^hi (simple arrow) and E/M (triangular arrow) cancer cells, and nuclear ZEB1^+ stromal cells (arrowheads) are also shown in the section stained with anti–E-cad and anti-ZEB1 antibodies (bottom right). **(D)** Kaplan–Meier survival analysis for high (red line) and low (black line) expression levels of the indicated genes in the described breast cancer patients. Hazard ratio (HR) is also shown. Data information: Asterisk indicates a significant difference relative to others (B) and the group with lower expression (D). Fisher's exact test (B) and Cox proportional hazards regression test (D). Scale bars, 30 μm (C-IHC) and 10 μm (A and C-IF). See also Fig S8 and Tables S2, S3, S4, and S5. ns, not significant; RFS, relapse-free survival.

Her2-amplified human breast cancers than in luminal A and basal-like breast cancers (Toullec et al, 2010). Inhibition of stromal SDF-1–dependent CXCR4 activation by CXCR4 inhibitors attenuates myofibroblast content, tumor angiogenesis, and tumor growth in patient-derived tumor xenografts from Her2^+ human breast cancers (Lefort et al, 2017). SDF-1 treatment also transactivates Her2 via CXCR4 present on cultured human breast and prostate carcinoma cells in a paracrine fashion (Cabioglu et al, 2005; Chinni et al, 2008). These series of findings indicate that Her2^+ cancer cells may be capable of generating SDF-1–producing CAFs that, in turn, influence tumor growth and progression, potentially resulting in poor outcomes in breast cancer patients. The TGF-β signal can also reportedly synergistically activate PI3K/AKT and Ras/MAPK pathways with Her2 signaling and accelerate the metastasis of Her2-derived mammary tumors in mice (Muraoka et al, 2003). Thus, stromal SDF-1 and TGF-β might play key roles in mediating the tumor-promoting interaction between the CAFs and Her2^+ER^−PR^− breast carcinoma cells. Molecular insights into how CAF-primed E^hi and E/M states are involved in Her2 expression on breast cancer cells also await further studies.

### CAF-primed tumor cell clusters resemble inflammatory breast cancer (IBC) emboli

IBC, which is rare and aggressive, is exemplified by Her2^+ and triple-negative breast cancer subtypes (Lim et al, 2018). Three-quarters of

IBC patients also develop tumor emboli, likely responsible for seeding lung metastases at a higher incidence and worsening the prognosis relative to non-IBC patients. IBC emboli are also characterized by increased cell–cell adhesion with high E-cad expression and the hybrid E/M state (Jolly et al, 2017). Notably, E-cad expression is required for IBC cells to form tumor emboli and disseminate into the lungs in an IBC xenograft murine model (Tomlinson et al, 2001). These findings indicate the functional resemblance of CAF-primed E^hi and E/M tumor cell clusters with IBC emboli, supporting the potential clinical relevance of the CAF-driven tumor cell clusters. Whether and, if so, how CAFs are involved in the pathogenesis and progression of IBC remains, however, to be determined.

Based on the clinical significance of CAF-promoted highly metastatic tumor cell clusters, therapeutic targeting of such tumor cell clusters and CAFs has enormous potential for developing treatments that efficiently prevent metastatic spread.

## Materials and Methods

### Reagents and cell lines

hTERT-immortalized, puromycin-resistant human mammary fibroblasts were subcutaneously injected with MCF-7-ras human

breast tumor cells into nude mice. The injected human mammary fibroblasts were then extracted from the developing tumor xeno-grafts for culture in the presence of puromycin (1 µg/ml) and the resulting puromycin-resistant cells were termed experimentally generated CAFs (exp-CAF2 cells) that acquired an activated myo-fibroblastic, tumor-promoting property, mediated by establishment of SDF-1 and TGF-β autocrine signaling during the course of tumor progression (Kojima et al, 2010; Polanska et al, 2011). The same parental human mammary fibroblasts that had been injected without tumor cells subcutaneously into mice were also extracted for culture in the presence of puromycin (1 µg/ml), and the resulting puromycin-resistant cells were termed control fibroblasts. CAFs and control fibroblasts were cultured in DMEM with GlutaMAX (Gibco) supplemented with 10% FBS (Sigma-Aldrich) and penicillin (100 U/ml)–streptomycin (100 µg/ml; Gibco). MCF10DCIS.com and MCF10CA1d cells were purchased from Asterand Bioscience and cultured in DMEM/F12 with GlutaMAX (Gibco) supplemented with 5% FBS and penicillin-streptomycin. DCIS cells extracted from tumor xenografts were also cultured in the same medium. MCF-7-ras cells (Orimo et al, 2005) were cultured in RPMI 1,640 medium with GlutaMAX (Gibco) supplemented with 10% FBS and penicillin–streptomycin. Recombinant human SDF-1 and TGF-β1 proteins were purchased from R&D systems. PP1 (Calbiochem) and saracatinib (Selleck) were also purchased.

### DNA construct

A pMX-c-Src (Y527F)-hygro retroviral vector, the active Src mutant (Kano et al, 2008), was kindly provided by Dr. Shigetsugu Hata-keyama, Hokkaido University. The tdTomato cDNA construct, kindly provided by Dr. Tsukasa Shibue (Whitehead Institute), was cloned into a pWZL-blast vector. The shRNA oligonucleotides against E-cad, CAM5, CAM6, ZEB1, and Src were generated and cloned into a lentivirus-derived pLKO1-hygro-shRNA vector. We also used pLKO1-hygro-CXCR4-shRNA and pLKO1-hygro-TβRII-shRNA vectors whose knockdown effect had been confirmed (Kojima et al, 2010). The target sequences are listed in Table S6.

### FACS analysis

We harvested 30-d-old tumor xenografts from mice, which had been subcutaneously injected with tdTomato-labelled DCIS cells plus either CAFs or control fibroblasts. The tumors were then minced and dissociated into single-cell suspensions by treatment with 1.5 mg/ml collagenase type I (Sigma-Aldrich) with gentle ag-itation for 2 h at 37°C.

To measure E-cad–positive DCIS cells, the single-cell suspen-sions were stained with anti–E-cad antibody (DECMA-1)-Alexa Fluor 488 for 40 min at 4°C. Dead cells were also eliminated by staining with 0.5 µg/ml DAPI (Thermo Fisher Scientific). E-cad positivity was then determined on tdTomato$^+$DAPI$^-$ DCIS cells to quantify E-cad$^+$ and E-cad$^-$ tumor cell proportions. The average of three in-dependent sets of data is also shown (Fig 1C).

To measure E-cad– and ZEB1-positive DCIS cells, the single-cell suspensions dissociated from tumors were stained with anti–E-cad antibody (DECMA-1)-Alexa Fluor 488 for 40 min at 4°C and then permeabilized with IntraPrep Permeabilization Reagent (BECKMAN

COULTER) before staining with anti-ZEB1-Alexa Fluor 647 (Novus Biologicals) for 30 min at 4°C. Positivity for both E-cad and ZEB1 was then determined on tdTomato$^+$ DCIS cells to quantify E-cadhiZE-B1lo/neg (Ehi), E-cadloZEB1hi (E/M), and E-cadhiZEB1hi cancer cell proportions (Fig S1G). The average of three independent sets of data is also shown (Fig 1D). The stained cells were detected with an LSRFortessa (BD Bioscience) and analyzed with FlowJo v10 software (Tree Star, Inc.).

### Animal studies

Male NOD/Shi-scid IL2 γ null (NOG) mice at 6 wk of age were purchased from the Central Institute for Experimental Animals (Kanagawa, Japan). The mice were bred under germ-free and specific pathogen-free conditions, and the experiments were ap-proved by the Animal Research Ethics Committee of the Juntendo Faculty of Medicine.

Subcutaneous injection was performed as previously described (Orimo et al, 2005; Kojima et al, 2010). Cancer cells and fibroblasts were harvested by trypsinization, washed once in PBS solution, and then re-suspended in DMEM with 5% FBS. Cell suspensions (400 µl) including 1 × 10$^5$ carcinoma cells and/or 3 × 10$^5$ fibroblasts with 50% Matrigel were then injected subcutaneously.

To examine lung colonization, cancer cells were harvested by trypsinization, washed once in PBS, and then resuspended in PBS with 1% FBS. Tumor cell suspensions (300 µl), including 5 × 10$^4$ carcinoma cells were then injected into the tail vein.

To examine metastatic colonization of various tissues, including bone and the liver, tumor cell suspensions (100 µl) including 1 × 10$^5$ GFP-labelled DCIS carcinoma cells were injected into the left ventricle of NOG mice as previously described (Hiraga & Nakamura, 2009). Metastatic lesions observed under fluorescent microscopy were confirmed by histological analysis.

### Measurement of metastases

To appropriately evaluate lung metastasis of carcinoma cells subcutaneously injected with or without fibroblasts into recipient mice, the metastasis index was calculated as the ratio of metastasis formation, as gauged by pulmonary nodule volume (Figs S1I, 2F, 4A–C, 7C, and F) and tdTomato fluorescent intensity (Fig 1F) in the lungs at 60 d after injection, relative to the weights (in grams) of the primary tumors resected at 30 d after injection (Figs S1H, 2C, and 4A–C). The sizes of lung metastatic nodules (0.5–5 mm in diameter) were measured using a scaler, and the tdTomato fluorescence intensity of metastatic nodules was measured under fluorescence dissecting microscopy.

Lung metastases formed by intravenous injection of carcinoma cells were also evaluated by measuring metastatic nodule sizes (Figs 2G, 4A–C, 6K, and S5J) and tdTomato fluorescent intensity (Figs 2H and 5E). Bone and liver metastases formed after intracardiac injection of GFP-labelled carcinoma cells were measured by GFP fluorescent intensity (Fig 2I). Fluorescence intensity was measured using a fluorescence stereomicroscope Leica M165 FC (Leica) and then quantitatively analyzed using cell profiler cell image analysis software (http://cellprofiler.org/).

## Preparation of blood smear

Peripheral blood (~500 $\mu$l) was collected from the submandibular veins of 4–6 mice bearing 30-d-old subcutaneous tumors (Fig 8A and B). Blood sampling was repeated on a weekly basis until the objective was achieved. Samples transferred into EDTA-containing tubes were incubated for 10 min with lysis buffer (150 mM $NH_4Cl$, 40 mM $KHCO_3$, and 0.1 mM EDTA) to eliminate red blood cells. The resulting cells were then centrifuged at 1,400 $g$ for 5 min and washed twice with PBS. The pellets were dissolved in 500 $\mu$l of 1% FBS in PBS. The 100-$\mu$l cell suspension was subsequently pasted to a silane-coated slide at 72 $g$ for 3 min using a Cytospin 4 Cyto-centrifuge (Thermo Fisher Scientific) to prepare, at a minimum, five slides per mouse. The slides were dried in air for more than 20 min and stored at −80°C.

## Colony-formation assay for CTCs

Peripheral blood (~500 $\mu$l) was collected from the submandibular veins of mice bearing 30-d-old subcutaneous tumors generated by DCIS cells expressing a pWZL-tdTomato-blast vector admixed with no fibroblasts (n = 9), control fibroblasts (n = 11) or CAFs (n = 11), as shown in Fig 8B. Red blood cells were eliminated, as described above. The cell pellets were dissolved with 5% FBS-DMEM/F12 before seeding on a six-well plate in the presence of blasticidin (1 $\mu$g/ml) for 21 d. The number of blasticidin-resistant, tdTomato+CTC colonies larger than 1 mm in diameter was quantified.

## Data availability

Microarray data for DCIS[cnt2cy] and DCIS[CAF2cy] were extracted from the Gene Expression Omnibus database under the GSE119253 accession number.

## Organoid culture

We seeded 1 × 10[5] DCIS cells onto 150 $\mu$l of Matrigel coating a 24-well plate, followed by culture with the DMEM/F12 medium with GlutaMAX supplement containing 5% FBS overnight (Figs 2C and S2A). Floating dead cells and media were removed the next day and viable cells attached to the Matrigel were covered with 50 $\mu$l of Matrigel and overlaid with media containing 5% FBS. The culture medium was changed every second day and the cells were cultured for 5 d. The Matrigel was mechanically destroyed and the organoids were collected into a 1.5-ml tube prior to being jellied using *iPGell* (GenoStaff) and fixed in formalin solution.

## Organotypic invasion assay

Organotypic invasion assay was performed as previously described with modifications (Ranftl & Calvo, 2017). First, 1 × 10[5] control fibroblasts or CAFs were embedded into 500 $\mu$l of the mixture of 2.3 mg/ml collagen gel (Corning) and 2.2 mg/ml Matrigel (Corning) that was adjusted to pH 7.0 with 0.1 M NaOH and then cultured on a 24-well culture plate with 1 ml of DMEM plus 10% FBS overnight (Fig 1E). Then, 2 × 10[5] DCIS cells were seeded onto the fibroblast-containing gel with 1 ml of a 1:1 mixture of DMEM and DMEM/F12 with 7.5% FBS (designated 1:1 medium) for 3 d. DCIS cells seeded onto the gel were next coated with 60 $\mu$l of Matrigel for 1 h and then overlaid with the 1:1 media. The gel was then mounted on a 100-$\mu$m cell strainer (BD Falcon) and fed from underneath with 9 ml of the 1:1 media on a six-well culture plate for 9 d before fixation with formalin. The media were changed every other day.

1 × 10[5] human mammary fibroblasts were embedded into 500 $\mu$l of the mixture of 2.3 mg/ml collagen gel (Corning) and 2.2 mg/ml Matrigel (Corning) that was adjusted to pH 7.0 with 0.1 M NaOH and then cultured on a 24-well plate with 1 ml of DMEM plus 10% FBS overnight (Fig 5D). 2 × 10[5] DCIS[cnt2cy] or DCIS[CAF2cy] expressing the indicated shRNA were then seeded onto the fibroblast-containing gel with 1 ml of the 1:1 medium for 3 d. DCIS cells seeded onto the gel were next coated with 60 $\mu$l of Matrigel for 1 h and then overlaid with the 1:1 media. The gel was then mounted on a 100-$\mu$m cell strainer (BD Falcon) and fed from underneath with 9 ml of the 1:1 media on a six-well culture plate for 9 d before fixation with formalin. Media were changed every other day. The numbers of tumor cell clusters were then counted using 400× magnification under a microscope in 10~12 fields in total, in each group (n = 3).

1 × 10[5] human mammary fibroblasts were embedded into 500 $\mu$l of the mixture of 2.3 mg/ml collagen gel (Corning) and 2.2 mg/ml Matrigel (Corning) that was adjusted to pH 7.0 with 0.1 M NaOH, followed by being cultured on a 24-well plate with 1 ml of DMEM plus 10% FBS overnight (Fig 7L). 2 × 10[5] DCIS cells were then seeded onto the fibroblast-containing gel with 1 ml of the 1:1 medium in the presence or absence of recombinant TGF-$\beta$1 (10 ng/ml) and SDF-1 (100 ng/ml) for 3 d. DCIS cells seeded onto the gel were next coated with 60 $\mu$l of Matrigel for 1 h and then overlaid with the 1:1 media. The gel was then mounted on a 100-$\mu$m cell strainer (BD Falcon) and fed from underneath with 9 ml of the 1:1 media with TGF-$\beta$1 (10 ng/ml) and SDF-1 (100 ng/ml) on a six-well culture plate for 9 d before fixation with formalin. The media were changed every other day.

## Measurement of apoptotic tumor cells in vitro

We seeded 2.5 × 10[3] DCIS[CAF2cy] expressing different shRNAs in non-FBS in DMEM/F12 on a low attachment 96-well plate. Apoptotic cell death was visualized and measured 6 h after seeding the cells using Caspase-3/7 Green Apoptosis Assay Reagent (Essen BioScience) and IncuCyte ZOOM (Essen BioScience). Caspase-3/7 activation was measured by counting the number of positive caspase-3/7 (green) objects and cell confluence (phase), as shown in Fig 4G.

## Cell proliferation assay

We seeded 5 × 10[3] DCIS[CAF2cy] or DCIS[cnt2cy] in DMEM/F12 with 5% FBS on a 96-well dish. The cells were treated with DMSO, saracatinib (1 $\mu$M) or PP1 (2 $\mu$M) for 60 h and cell growth was measured as cell confluence (phase) using IncuCyte ZOOM, as shown in Fig 6G.

## Quantification of cPARP[+] or Ki-67[+] cells

Sections were prepared from experimental lung metastases generated by intravenous injection of DCIS[CAF2cy] expressing GFP-, E-cad−, CAM5- or CAM6-shRNAs (n = 4). To detect apoptotic and

proliferating cells, these sections were stained with anti-cPARP and anti–Ki-67 antibodies, respectively. The numbers of tumor cells positive or negative for cPARP or Ki-67 were then counted using 400× magnification under a microscope in 11–32 metastatic nodules in total, in each group (n = 4). Ratios of the positive cell number per the total cell number were calculated before being denoted as one in DCIS$^{CAF2cy}$ expressing GFP-shRNA (Fig 4H).

### Cell-to-cell adhesion assay

$1 \times 10^5$ DCIS$^{alone2cy}$, DCIS$^{cnt2cy}$, or DCIS$^{CAF2cy}$ were seeded in DMEM/F12 with 5% FBS as layer cells on an eight-well glass plate for 24 h. Then, $2 \times 10^4$ GFP-positive DCIS$^{alone2cy}$, DCIS$^{cnt2cy}$, or DCIS$^{CAF2cy}$ were seeded as adherent cells on the layer cells for 20 min before fixation with 4% PFA. The GFP-positive cell number was counted on each well and four wells per group were analyzed (Figs 4D and S4D). DCIS$^{cnt2cy}$ and DCIS$^{CAF2cy}$ expressing GFP-, E-cad–, CAM5-, or CAM6-shRNA were also seeded as layer cells before seeding of GFP-positive DCIS$^{CAF2cy}$ as adherent cells on top of the layer cells, as shown in Fig 4D. DCIS$^{cnt2cy}$ and DCIS$^{CAF2cy}$ treated with DMSO, saracatinib (1 μM), or PP1 (10 μM) were also seeded as layer cells for 24 h before seeding of GFP-positive DCIS$^{CAF2cy}$ as adherent cells on top of the layer cells in Fig 6I.

### Cell-to-cell aggregation assay

First, $1 \times 10^4$ DCIS$^{alone2cy}$, DCIS$^{cnt2cy}$, or DCIS$^{CAF2cy}$ suspended in DMEM/F12 with 5% FBS were seeded onto a low attachment 96-well plate for 24 h, as shown in Fig 4E (left). DCIS$^{cnt2cy}$ and DCIS$^{CAF2cy}$ expressing GFP-, E-cad–, CAM5-, or CAM6-shRNA were also seeded onto the low attachment 96-well plate for 24 h (Fig 4E, right). We also seeded DCIS$^{CAF2cy}$ onto the low attachment 96-well plate in the presence of DMSO, saracatinib (1 μM), or PP1 (10 μM) for 24 h, as shown in Fig 6J. To estimate the volume (mm$^3$) of cell aggregates, major and minor axes of each aggregate were measured in a total of 20–26 aggregates in five to eight wells per group using Aqua Cosmos 2.6 software (Hamamatsu Photonics). The cell aggregate volume was denoted as one in DCIS$^{CAF2cy}$ treated with GFP-shRNA (Fig 4E) or DMSO (Fig 6J).

### ECM–cell adhesion assay

The 96-well plates were coated with collagen I solutions (50 μg/ml) at 37°C for 1 h, followed by washing twice using DMEM with 1% BSA. Tumor cell suspensions (100 μl) including $2 \times 10^4$ DCIS cells treated with the indicated shRNA or inhibitors were seeded on the plate for 1 h in a 37°C CO$_2$ incubator followed by shaking the plate (AS ONE) at 600 $g$ for 15 s and then washing with PBS twice, as shown in Fig 4F. After fixation of the cells with 4% PFA for 10 min, they were stained with crystal violet for 10 min, washed, and then completely dried. Finally, 33% acetic acid was added for 20–30 min before optical density measurement at 550 nm using a microplate reader (Bio-Rad).

### Scratched wound invasion assay

First, a 50-μl quantity of Matrigel (100 μg/ml, BD Bioscience) was added onto an Image Lock 96-well plate (Essen BioScience)

overnight at 37°C in 5% CO$_2$. Then, $6 \times 10^4$ DCIS$^{alone2cy}$, DCIS$^{cnt2cy}$, and DCIS$^{CAF2cy}$ were seeded onto the Matrigel-precoated plates in 100 μl of DMEM/F12 with 5% FBS for 4 h at 37°C in 5% CO$_2$ (Figs 2B and 5C). The scratch wound was also generated by the Wound Maker tool (Essen BioScience), before washing with PBS once and coating with 50 μl of Matrigel (8 mg/ml) on a prechilled cool box. To solidify the Matrigel, the plate was subsequently incubated at 37°C in 5% CO$_2$ for 30 min followed by addition of 100 μl of 5% FBS in DMEM/F12 at 37°C in 5% CO$_2$. The plate was scanned with an IncuCyte Zoom (Essen BioScience) according to the manufacturer's instructions at 3-h intervals for 5 d.

Plates were incubated for 1 h and then treated with DMSO, saracatinib (1 μM), or PP1 (10 μM) (Fig 6H). The plates were scanned with an IncuCyte Zoom (Essen BioScience) according to the manufacturer's instructions at 2-h intervals for 4 d.

### Generation of GFP-labelled tumor cells

Lentivirus infection of DCIS$^{CAF2cy}$ was performed using a PRRL-GFP vector as described previously (Onder et al, 2008). The resulting cells were sorted by GFP positivity using the MoFlo Astrios cell sorter (Beckman Coulter) before their intracardiac injection into recipient NOG mice (Fig 2I).

### Detection of GFP and tdTomato fluorescence

The 21-d-old tumor xenografts generated by tdTomato-labelled DCIS cells admixed with GFP-expressing fibroblasts were dissected from mice and fixed overnight in 4% PFA in PBS at 4°C (Fig S1B). Tumors were then cryopreserved using 30% sucrose in PBS overnight at 4°C and embedded in optimal cutting temperature (OCT) compound for cryosectioning. Frozen sections were washed with PBS and counterstained with DAPI before visualization under a fluorescence Axioplan 2 microscope (Zeiss).

### Purification of rat IgG against E-cadherin

Hybridoma DECMA-1 (anti–E-cadherin, rat IgG1) (Vestweber & Kemler, 1985) was cultured in HL1 medium (Lonza Walkersville, Inc.). Rat IgG was purified from the culture media by Protein G Sepharose 4 Fast Flow (GE Healthcare) before conjugation with Alexa Fluor 488 using an Alexa Fluor monoclonal antibody labeling kit (Thermo Fisher Scientific).

### Western blot analysis

The whole-cell lysate was prepared using SDS gel loading buffer. The lysate was separated by SDS–polyacrylamide gel electrophoresis (8% acrylamide) and transferred to PVDF membranes. Primary antibodies were incubated in TBS-T (0.05 M Tris-buffered saline and 0.05% Tween 20) with 10% FBS after blocking with 5% skim milk. EnVision+ System-HRP Labelled Polymer antimouse or antirabbit antibodies and polyclonal rabbit antigoat immunoglobulins/HRP (DAKO) were used as secondary antibodies. Detection was performed using a ChemiDoc MP System (Bio-Rad Laboratories) with HRP chemiluminescence substrate. Signal intensity was also measured using Image Lab Software.

### Retroviral and lentiviral infections

Retroviral and lentiviral infections were performed as described previously (Stewart et al, 2003). After infection, the cells were cultured for 4–6 d in the presence of the appropriate antibiotic for each plasmid; blasticidin (1 µg/ml), puromycin (1 µg/ml), or hygromycin (50 µg/ml).

### Real-time PCR

Real-time PCR was performed as previously described (Kojima et al, 2010). Total RNA was extracted using NucleoSpin RNA II (Takara) in accordance with the manufacturer's protocol. SuperScript II reverse transcriptase (Invitrogen) was also used to synthesize cDNA. The resulting cDNAs were used for PCR using Fast SYBR Green Master Mix (Applied Biosystems) in triplicate. PCR and data collection were performed with a 7500 Fast Real-Time PCR System (Applied Biosystems). Relative gene expressions were analyzed by the ΔΔCt method. Data were normalized relative to the GAPDH or $\beta$-actin gene expression level. The primer sequences are listed in Table S6.

### High-throughput screening

The chemical library including 358 chemical compounds was kindly provided by the Screening Committee of Anticancer Drugs supported by a Grant-in-Aid for Scientific Research on Innovative Areas, Scientific Support Programs for Cancer Research, from The Ministry of Education, Culture, Sports, Science and Technology, Japan. We seeded $1 \times 10^5$ DCIS[CAF2cy]/well on 24-well culture plates and then treated them with each compound for 24 h before isolation of RNA and real-time PCR for CAM6 expression (Fig S5A).

## Microarray Data Collection

Amplified double-stranded cDNA was generated using the WT-Ovation Pico RNA Amplification System (Cat. No. 3300-12; NuGEN) and biotin labelled using the FL-Ovation cDNA Biotin Module v2 (Cat. No. 4200-60; NuGEN) according to the manufacturer's instructions. Overnight hybridization to GeneChip Human Genome U133 Plus 2.0 Arrays (Cat. No. 900466; Affymetrix) was performed according to the NuGEN FL-Ovation cDNA Biotin Module v2 user guide. Arrays were washed and stained using the Affymetrix GeneChip Fluidics Station and scanned using the Affymetrix GeneChip Scanner 3000 system with Autoloader, running on Affymetrix GeneChip Command Console Software.

### Immunohistochemistry

The use of formalin-fixed paraffin-embedded (FFPE) tissue specimens of breast cancer in this study was approved by the Juntendo University ethics review board. FFPE invasive breast carcinomas were prepared from breast cancer patients who had received neither preoperative chemotherapy nor hormone therapy (Fig 9A and C). We also used FFPE tissue specimens prepared from primary tumors and metastases dissected from mice bearing tumors. The 3-µm-thick sections were prepared from FFPE tissue specimens and deparaffinized. The slides were then treated with 0.3% $H_2O_2$ in methanol for 20 min at room temperature. Antigen retrieval was also performed by autoclaving in 10 mM citrate buffer at pH 6.0 for 20 min at 120°C. Moreover, the slides were incubated with primary antibody at 4°C overnight. EnVision+ System-HRP Labelled Polymer (DAKO) or histo-fine simple stain MAX-PO (G) (Nichirei Bioscience) was used as the secondary antibody for mouse or goat IgG, respectively. The incubation was carried out for an hour at room temperature. Diaminobenzidine was used as a chromogen followed by hematoxylin counterstaining. Antibodies used are listed in Table S7.

### Immunofluorescence

FFPE tissue sections were prepared from DCIS tumor xenografts, DCIS cells in/on gel, and human breast tumor tissues (Fig 9A and C). The slides were subjected to antigen retrieval, blocking with 5% BSA in PBS and then incubated with both anti–E-cad and anti-ZEB1 antibodies at 4°C overnight.

Also, $5 \times 10^4$ DCIS[CAF2cy] and DCIS[cnt2cy] were seeded onto a silane-coated slide at 72 $g$ for 3 min using Cytospin 4 Cytocentrifuge (Thermo Fisher Scientific) followed by air-drying for 20 min and stored at –80°C. The slides were fixed with 4% PFA for 10 min and washed twice with PBS and fixed in methanol for 5 min and again washed twice with PBS. The slides were then blocked in blocking solution at 37°C for 30 min before incubation overnight at 4°C with dilution of the primary antibodies including anti–E-cad, anti-CAM5, anti-CAM6, and anti-Src antibodies, as shown in Figs 3G, 6E, and S5G. $3 \times 10^4$ cancer cells were also seeded onto eight-well chamber slides (Thermo Fisher Scientific) before staining with anti–E-cad or anti-ZEB1 antibody in Figs S2F and S5E (right), Figs 6F, S6C, 7D, and H.

To prepare frozen sections, tumors raised by DCIS[CAF2cy] and DCIS[cnt2cy] subcutaneously injected into mice were fixed overnight in 4% PFA in PBS at 4°C. Tumors were then cryoprotected using 30% sucrose in PBS overnight at 4°C and embedded in OCT for cryosectioning before staining with anti–p-Src antibody, as shown in Fig S5B. The slides were then incubated with the corresponding fluorescence-conjugated secondary antibodies (Thermo Fisher Scientific) for an hour at room temperature. After being stained with DAPI, the tissues were observed under a fluorescence Axioplan 2 microscope (Zeiss). Images were also acquired with a CCD camera and prepared using the ImageJ image analysis program. Antibodies used are listed in Table S7.

### Tissue microarray

Tumor tissue microarrays were constructed with 257 formalin-fixed primary breast cancers, as reported previously (Daigo & Nakamura, 2008). The tissue area for sampling was selected based on visual alignment with the corresponding hematoxylin and eosin–stained section on a slide. Several tissue cores (diameter 0.6 mm; height 3–4 mm) taken from a donor tumor block were placed into a recipient paraffin block using a tissue microarrayer (Beecher Instruments). A core of normal tissue was punched from each specimen, and 5-µm sections of the resulting microarray block were used for immunohistochemical analysis.

To investigate the presence of CAM5, CAM6, E-cad, and ZEB1 proteins in clinical samples that had been embedded in paraffin blocks, the sections were stained in the following manner, as shown in Fig 9B and Tables S2, S3, S4, and S5. First, the antibody was added after blocking of endogenous peroxidase. The sections were incubated with HRP-labeled antigoat immunoglobulin G as the secondary antibody. Substrate chromogen was added, and the specimens were counterstained with hematoxylin. On immunohistochemical analyses, we confirmed the specificity of the antibodies used by immunoblotting using DCIS cells overexpressing E-cad, CAM5, and/or CAM6 as well as DCIS$^{CAF2cy}$ expressing shRNAs against E-cad, CAM5, CAM6, or ZEB1. Three independent investigators assessed the results semi-quantitatively without prior knowledge of the clinicopathological data. Whether staining was positive or negative for CAM5/CAM6/E-cad or E-cad/ZEB1 was evaluated in tumor cells. Cases were taken to be positive if two or more investigators independently defined them as such. Her2$^+$ tumors are those with 3+ and 2+ Her2 staining according to the ASCO guideline (Wolff et al, 2007).

### Analysis of DNA methylation by pyrosequencing

Pyrosequencing was carried out as described previously (Sugai et al, 2017). The DNA methylation status of each gene promoter region was investigated by PCR of bisulfite-modified genomic DNA (EpiTect Bisulfite Kit; QIAGEN) using pyrosequencing for quantitative methylation analysis (Pyromark Q24; QIAGEN), as shown in Fig S9A. The primers were designed using the Pyromark Assay Designing Software (QIAGEN), with 3–4 CpG sites included in the analysis of promoter methylation. For bisulfite sequencing, amplified PCR products were cloned into a pCR2.1-TOPO vector (Thermo Fisher Scientific), and 10–15 clones from each sample were sequenced using an ABI3130x automated sequencer (Thermo Fisher Scientific). Primer sequences and PCR product sizes are listed (Table S6).

### In situ PLA

In situ PLA was performed according to the protocol of Duolink In Situ/Fluorescence (Sigma-Aldrich), as shown in Fig 3G (third from the top), Figs S3E, S5F, and 6E (lower panels). We seeded 5 × 10$^4$ DCIS$^{CAF2cy}$ or DCIS$^{cnt2cy}$ on a silane-coated slide at 72$g$ for 3 min using Cytospin 4 Cytocentrifuge (Thermo Fisher Scientific) followed by air drying for 20 min and storage at –80°C. The slides were fixed with 4% PFA for 10 min and washed with PBS twice and fixed in methanol for 5 min and then again washed with PBS twice. The slides were then blocked in blocking solution at 37°C for 30 min before incubation with primary antibodies at 1:100 dilution overnight at 4°C. After washing with 1× buffer A for 5 min twice, PLA proximity probes including both Duolink PLA Anti-Rabbit MINUS and PLA Anti-Mouse PLUS (diluted) were added to samples and incubated at 37°C for 1 h. After washing with 1 × buffer A twice for 5 min, the diluted ligation solution, including ligase was added to samples and incubated at 37°C for 30 min. After washing twice with 1× buffer A for 5 min, the diluted amplification solution including polymerase was added at 37°C for 100 min. After washing twice with 1× buffer B for 10 min and washing with 0.01× buffer B for 1 min, the samples were mounted with Duolink In situ Mounting Medium with

DAPI and covered with a slide glass. Fluorescence was detected by a Zeiss Axioplan 2 fluorescent microscope (Zeiss). Images were acquired with a CCD camera and prepared using Adobe Photoshop CS5 software. Total fluorescent intensity in each group was analyzed using cell profiler cell image analysis software (http://cellprofiler.org/).

### Statistical analysis

Data distribution and the significance of differences between groups were analyzed by the two-tailed Mann–Whitney $U$-test, Wilcoxon rank sum test, and $t$ test. $P < 0.05$ was considered to indicate a statistically significant result.

### Linear correlation analysis

The Pearson product-moment correlation coefficient was used to measure the strength and direction of the linear association between two variables, as shown in Figs 3F and S3D (CAM6 and CAM5; E-cad and CAM5; and E-cad and CAM6). In this analysis, we used "cor.test" and "pt" in R programming language for computing the correlation coefficient and the $P$-value, respectively.

Correlations between CAM5, CAM6 and E-cad mRNA expressions or between E-cad and ZEB1 expressions were also investigated using the human breast metastasis dataset GSE14020, as shown in Fig 8E. The statistical significance of a correlation coefficient was evaluated by the two-sample correlation test using R statistical software (version 3.4.3, https://www.r-project.org/).

### Kaplan–Meier survival analysis

For each primary breast cancer dataset, the gene expression values of each sample were log2-scaled and then mean-centered. Subsequently, z-score transformation was performed to normalize gene expressions across all samples in each dataset. DNA microarray results were analyzed with R programming language. Class comparisons between DCIS$^{CAF2cy}$ and DCIS$^{cnt2cy}$ were performed to identify gene expression changes exceeding twofold and with a value of $P < 0.05$. We obtained a list of 44 genes as CIMS, as shown in Table S1. This gene set was used to classify samples from human primary breast cancer datasets (GSE7390, GSE12276, and GSE14333) (Figs 3A and S3A). Univariate Cox proportional hazards regression was used to identify genes significantly associated ($P$-value < 0.05) with distant metastasis-free survival and lung metastasis-free survival to estimate the regression coefficients between these differentially expressed genes. Kaplan–Meier survival analysis and log rank tests were performed using the survival package 2.37 in R 3.03 to compare the survival curves of low-risk and high-risk groups.

Kaplan–Meier survival analysis of the indicated genes in breast cancer patients was performed using a publicly available database (KM plotter; www.kmplot.com) by multivariate analysis using Cox proportional hazards regression, as shown in Figs S8A and 9D. The online resource for breast cancer DNA microarray analysis is available at http://kmplot.com/analysis.

### Gene set enrichment analysis (GSEA)

GSEA is widely applied, using a list of gene sets, to determine whether a predefined gene set shows a statistically significant difference between two biological states. We applied GSEA to up-regulated genes in DCIS$^{CAF2cy}$ relative to those in DCIS$^{cnt2cy}$ using MiSigDB (http://www.broadinstitute.org/gsea/msigdb/) in Fig 3B, where the normalized enrichment score and *P*-value are shown.

# Supplementary Information

# Acknowledgements

We thank S Hatakeyama for the active Src cDNA construct, T Shibue for the tdTomato cDNA construct, Drs. RA Weinberg and M Kasai for reviewing the manuscript and giving critical comments and UM Polanska, N Kadowaki, K Miyahara, K Shimizu, T Takagaki, C Kataoka, T Kobayashi, K Kajino, Y Ono, and members of the laboratories of A Orimo, K Okumura, and O Hino for technical and general assistance. We also thank the Screening Committee for Anti-cancer Drugs supported by a Grant-in-Aid for Scientific Research on Innovative Areas, Scientific Support Programs for Cancer Research, from The Ministry of Education, Culture, Sports, Science and Technology, Japan, for chemical library support. This work was supported in part by a Grant-in-Aid for Scientific Research on Innovative Areas from The Japan Society for the Promotion of Science (JSPS KAKENHI grant number JP: 16H06277). Funding for this work was also provided by the Juntendo University Young Investigator Award (Y Matsumura), the Juntendo University Joint Project Award (Y Matsumura), and Cancer Research UK Grant C147/A6058 (to A Orimo), Grants in-Aid for Scientific Research from the Ministry of Education, Culture, Sports, Science and Technology, Japan (24300332, 25640069, and 15K14385 to A Orimo) and a Grant-in-Aid (S1311011) from the Foundation of Strategic Research Projects in Private Universities from the MEXT, Japan (A Orimo), and the Juntendo University School of Medicine, Research Institute for Diseases of Old Age (A Orimo).

## Authors Contributions

Y Matsumura: formal analysis and investigation.
Y Ito: formal analysis and investigation.
Y Mezawa: formal analysis and investigation.
K Sulidan: formal analysis and investigation.
Y Daigo: formal analysis and investigation.
T Hiraga: formal analysis and investigation.
K Mogushi: data curation, formal analysis, and investigation.
N Wali: formal analysis and investigation.
H Suzuki: formal analysis and investigation.
T Itoh: formal analysis and investigation.
Y Miyagi: data curation, formal analysis, and investigation.
T Yokose: data curation, formal analysis, and investigation.
S Shimizu: data curation, formal analysis, and investigation.
A Takano: data curation, formal analysis, and investigation.
Y Terao: supervision.
H Saeki: data curation.
M Ozawa: resources.
M Abe: formal analysis and investigation.
S Takeda: supervision.
K Okumura: supervision and funding acquisition.
S Habu: resources and supervision.
O Hino: supervision.
K Takeda: formal analysis and investigation.
M Hamada: data curation, formal analysis, and investigation.
A Orimo: conceptualization, formal analysis, supervision, funding acquisition, validation, investigation, project administration, and writing—original draft, review, and editing.

## Conflict of Interest Statement

The authors declare that they have no conflict of interest.

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
