## [Reviewer comments · Life Science Alliance]

Life Science Alliance

Stromal fibroblasts induce metastatic tumor cell clusters via epithelial-mesenchymal plasticity

Yuko Matsumura, Yasuhiko Ito, Yoshihiro Mezawa, Kaidiliayi Sulidan, Yataro Daigo, Toru Hiraga, Kaoru Mogushi, Nadila Wali, Hiromu Suzuki, Takumi Itoh, Yohei Miyagi, Tomoyuki Yokose, Satoru Shimizu, Atsushi Takano, Yasuhisa Terao, Harumi Saeki, Masayuki Ozawa, Masaaki Abe, Satoru Takeda, Ko Okumura, Sonoko Habu, O Hino, Kazuyoshi Takeda, Michiaki Hamada, and Akira Orimo
DOI: <https://doi.org/10.26508/lsa.201900425>

Corresponding author(s): Akira Orimo, Graduate School of Medicine, Juntendo University

Review Timeline:	Submission Date:	2019-05-11
	Editorial Decision:	2019-05-17
	Revision Received:	2019-07-02
	Editorial Decision:	2019-07-04
	Revision Received:	2019-07-10
	Accepted:	2019-07-10

Scientific Editor: Andrea Leibfried

Transaction Report:

Please note that the manuscript was previously reviewed at another journal and the reports were taken into account in the decision-making process at Life Science Alliance.

Referee #1 Review

Matsumura et al reported their findings that carcinoma-associated fibroblasts (CAFs) induced the development of two distinct cancer cell populations that differ in expression levels of E-cadherin and ZEB1. They provided evidence that interaction with CAF leads to more aggressive tumor progression and metastasis to multiple organs. Molecularly, there are a number of molecules proposed to regulate this process and control metastasis efficiency, including adhesion molecules CEACAM5, CEACAM6, and E-cadherin, tyrosine kinase Src, and cytokines SDF1 and TGF β . Perturbations to these molecules abolished metastasis. At the technical level, most experiments are well performed and rigorously controlled. The conclusion is of potential interest but is weakened by multiple caveats as elaborated below.

First, the entire study is based on one cell line that is necessarily representative of the heterogeneous breast cancer. At least one other model should be utilized to support some of the key conclusions. Second, co-injection of CAFs and cancer cells subcutaneously is a suboptimal approach. Tumor evolution can be influenced by surrounding tissue environment. Mammary fat pad injection, or better yet, intraductal injection should be employed to validate some of the key conclusions. Third, the authors showed that knockdown of CEACAM5/6 led to E-cadherin reduction, and vice versa, indicating that these molecules mutually regulate each other. This may need to be understood better. The authors suggest that reduction of one of them destabilizes the others. This is hard to imagine unless CEACAM5/6 directly interact with E-cadherin. This needs to be examined and perhaps delineation of the kinetics of the process will support this conclusion better.

Fourth, is the phenotype truly reflecting cell plasticity or genetic heterogeneity? If the Ehi or E/M cells are sorted out and cultured *in vitro* separately, do they revert back to the original state? What about *in vivo*? This will also answer whether the changes are at the genetic level (implying some cell Darwinian selection) or at the epigenetic level. Along the same lines, does the maintenance of this phenotype require the constant presence of CAFs?

Fifth, based on conventional knowledge, it is conceivable that TGF- β leads to partial EMT... but the induction of Ehi population is particularly interesting. It would strengthen the manuscript if there is at least one experiment looking specifically at whether the presence of this Ehi population is altered by inhibition of Src/ TGF- β in DCIS-CAF1cy tumors *in vivo*.

Finally, it is difficult to understand why clinical association is only strong in Her2+ breast cancer patients - despite the fact all biology is studied in a "triple negative" model. This prompts, at the very least, the usage of a Her2+ model to bridge experimental and clinical data.

Referee #3 Review

In this manuscript, the authors propose a mechanism of tumor and stromal fibroblast (CAF) crosstalk, which favors metastatic spreading of mammary cancer cells. Using MCF10DCIS.com cells (DCIS) injected in the flank of NOG mice alone or admixed with either fibroblasts and with carcinoma-associated fibroblasts (CAF), the authors aim at demonstrating the existence of two distinct tumor cell populations within the primary tumor. They describe two populations, based on the expression of the epithelial marker E-cadherin and a mesenchymal transcription factor ZEB1. From the two populations that co-exist, one is E-cad high and ZEB1 low (called Ehi) and one is E-cad low and ZEB1 high (called E/M). These two populations strongly interact through cell-cell junctions within the tumor but also, within collective strands or clusters of invasive DCIS cells *in vivo*. The authors claim that CAF cells induce a partial epithelial to mesenchymal transition within the tumor mass responsible for metastatic spreading of the parental epithelial cancer cells. The authors argue that CAF promote the partial EMT through the paracrine secretion of both TGF β 1 and SDF1 cytokines (previously demonstrated by the team, this crosstalk is responsible for pro-tumorigenic activities of the CAF in a paracrine manner; Kojima et al., PNAS 2010).

In response to the paracrine signaling, the DCIS Eh cells express CEACAM 5 and 6 which are experimentally important for the metastatic spreading in the lung. The authors conclude that DCIS cells admixed with CAF (for two passages in vivo, called DCIScaf2cy) cells in vivo present an elevated level of active Src kinase which favors collective invasion. Experimental inhibition of cell-cell junction molecules (CEACAM 5 and 6) or inhibition of Src (expression or activity) result in decrease metastatic ability of the DCIS cells. Finally, the authors corroborate their experimental findings with human breast cancer either via an immunohistology analysis of a cohort of 257 patient samples and with publically available omics data from breast cancer patients.

Overall, this is an extensive study using both in vivo and in vitro models to demonstrate the significance of the two co-existing cell populations (Eh and E/M) promoted by the CAF secretome within the DCIS parental population. The authors deserve much credits for the multiple experiments presented in the manuscript. One issue with the presentation of these data is that, while the mechanistic connections are technically novel with regard to this specific model system, most of the elements have been found previously or could have been reasonably anticipated (e.g. CAF promotes tumor growth and metastasis; CAF supports collective invasion; CAF promotes cancer cell heterogeneity; partial EMT; Src function in cancer cell invasion). Nevertheless, they are important novelties emphasized in this manuscript such as the role of CEACAM proteins in cancer cell motility and metastatic spreading and CAF promotes circulating cancer cell clusters. Addition of correlative analysis from human samples is also an important aspect of this work. However, this work has a number of notable experimental caveats, and overall falls far short of presenting convincing evidence that would support the existence of such functional cooperation.

Major concerns

- The in vivo experimental model presented in this manuscript allows "continuous interaction between the injected tumor cells and human fibroblasts within a tumor mass" (page 6). To this respect, it is well established and accepted that the continuous interaction between tumor cells and surrounding fibroblasts lead to cells co-education and result in increase tumorigenic and metastatic abilities. It is also well established that a large number of resident fibroblasts are activated by the tumor cells and become activated into a carcinoma associated fibroblast (CAF) cells. In this manuscript, the authors used an admixed tumor mass, injected into NOG mice, constituted of DCIS tumor cells "barely metastatic in vivo non-invasive breast ductal adenocarcinoma MCF10DCIS.com (DCIS)" (page 6) and fibroblasts or CAF or DCIS cells injected alone (previously isolated and reported). Here, the authors show that the tumorigenic properties of the DCIS cells are supported specifically by the admixed with the CAF but not with the control fibroblasts or when injected alone. However, the authors failed to demonstrate that the DCIS cells used in this manuscript are unable to activate either the control fibroblasts admixed within the NOG mice nor to activate the host fibroblasts during the time of the experiment.

While it is clear that the DCIS-caf2cy cells (selected for their CAF-induced tumorigenic properties) are more potent to colonize the lung (SuppFig S2D and S2E) there is any doubt that the DCIS-alone1cy; DCIS-alone2cy; DCIS-cnt1cy and the DCIS-cnt2cy are also highly potent to colonize the lung (SuppFig S2D and S2E). Thus, it seems that the metastatic capacity of the DCIS is correlated

with the CAF crosstalk but not entirely dependent. Also, the ability of the parental cells to colonize the lung is missing in this set of experiments, which make difficult to reach a conclusion.

Next, I found that the authors failed to demonstrate that CAF trigger two clear cell populations from the DCIS parental cell. Indeed, DCIS admixed with CAF present cells that show high level of ZEB1 staining (Figures 1A, 1B, S1E, 2E and S2E). However, I found difficult to understand why the E-cadherin level of the DCIS admixed with control fibroblasts or injected alone, present such a low level of E-cadherin staining (Figure 1B, S1E, 2E and S2A). The FACs analysis presented in Figure S1F suggest a mixed population of cells instead of two distinct cell populations within the tumor mass. Also, immunohistological analysis of E-cadherin detection presented in Figure 1A (DCIS-cnt fibro.) shows a high level of E-cadherin expression, which is not present in all the IF shown in the manuscript.

Altogether, I Would like to emphasize much of the credits that the authors deserve for this interesting study, but I think that a more robust direct evidence should be made on the role of CAF in the appearance of the two distinct cell populations in the DCIS model. I found that most of the evidences presented here are indirect and some statement are more correlative then basic scientific demonstration.

Other major concerns need to be addressed:

- To my point of view, a more direct evidence that the DCIS E/M cells provide leading collective cancer cell strand, would be to show, using organotypic culture assays (i.e. F1E) that the DCIS E/M cells (sorted by FACs) could invade independently of CAF. Indeed, in this model, CAF are the leading cells of the invading cluster. So, if DCIS E/M cells could promote collective invasion, they should be able to replace CAF cells in this assay.

To this respect, what is the mesenchymal phenotype of the DCIS E/M cells (in regards to the DCIS Ehi)? What is the level of expression of mesenchymal-related transcription factor other than ZEB1? Do they secrete MMPs? Are they contractile?

Also, a clear demonstration of the importance of the two distinct cell populations in cancer growth and metastatic spreading would be obtained by independent sub-cutaneous injection of the DCIS Ehi and DCIS E/M after cell sorting.

The set of experiments obtained by multiple cell injection in vivo and isolation of the DCIS caf2cy (and controls) is an important aspect of this manuscript. It seems that cells retain the memory of the co-injection even after in vitro culture. It is stated in the discussion that epigenetic events seem not to be a crucial mechanism here. Therefore, how would you explain the long lasting effect observe using those cells?

Since DCIS cells are barely not metastatic in vivo, and their pro-metastatic capacity is induced by the CAF through the appearance of the two distinct cell populations. What is the statue of the Ehi and E/M populations in a more aggressive breast cancer model (i.e. human MDA-MB-231 cells or others?). Are they all E/M state?

What is the level of TGFb1 and SDF1 secretion in the DCIS cells? One would consider that DCIS could eventually secrete moderate level of these two cytokines, and therefore an autocrine loop

could potentially take place, and therefore ruled out the hypothesis of the paper, since the DCIS alone does not show high level of DCIS E/M phenotype.

At experimental level, DCIS caf2cy cells show a higher level of CAM5 (more than 100 fold of induction compare to ctrl cells) compared to CAM6 (around 40 fold) (Fig 3D), while, the TGFb1 and SDF1 combo of stimulation of DCIS cells in vitro induced more CAM6 than CAM5 at low rate (7 and 1.5 fold respectively). How the authors explain such a discrepancy, would it mean that many other mechanisms regulate CAM5 and 6 expressions in DCIS?

Finally, the concept involving CAF and DCIS cells crosstalk for CTC clusters formation within the systemic circulation, as suggested by the schematic cartoon presented in F8F seems to not be supported enough in this work by experimental data. This concept would deserve a full manuscript of investigation by itself.

May 17, 2019

Re: Life Science Alliance manuscript #LSA-2019-00425-T

Dr. Akira Orimo
Graduate School of Medicine, Juntendo University
Department of Molecular Pathogenesis
2-1-1, Hongo
Bunkyo-ku, Tokyo 113-8421
JAPAN

Dear Dr. Orimo,

Thank you for submitting your manuscript entitled "Stromal fibroblasts induce metastatic tumor cell clusters via epithelial-mesenchymal plasticity" to Life Science Alliance. The manuscript was assessed by expert reviewers at another journal before and the editors transferred those comments to us with your permission.

The reviewers appreciated the effort that went into the analysis, but thought that more definitive support for the core claim on induction of differential EMT states by CAFs is lacking. We still think that the observations made are valuable, and we would thus like to invite you to provide a revised version for publication in Life Science Alliance. We expect a point-by-point response and accordingly changes to the manuscript text and discussion, acknowledging where future work will provide more definitive insight and support for the conclusions drawn. Furthermore, the following editorial points should get addressed:

- In line with our general policies, we would like to display minimally processed raw data alongside the final figures for both Fig 1A and 2D, but ideally also for the other figures
- please indicate in the figure legend to Fig. S1K that the same image is shown as in Fig. 1G.
- note that we only have supplementary figures and tables (not appendix figures/tables) in LSA => please re-name accordingly
- please upload all figures and supplementary figures as individual files, the suppl. Figure legends can go into the main manuscript text as well
- please incorporate the appendix methods into the main manuscript
- the tables should all get provided as word docx or excel files, please
- note that a callout to table S7 is currently missing in the text
- please increase the scale bar for the figures, they are rather small in many of them.

While you are revising your manuscript, please also attend to the below editorial points to help

expedite the publication of your manuscript. Please direct any editorial questions to the journal office.

Thank you for this interesting contribution to Life Science Alliance. We are looking forward to receiving your revised manuscript.

Sincerely,

B. MANUSCRIPT ORGANIZATION AND FORMATTING:

We encourage our authors to provide original source data, particularly uncropped/-processed

electrophoretic blots and spreadsheets for the main figures of the manuscript. If you would like to add source data, we would welcome one PDF/Excel-file per figure for this information. These files will be linked online as supplementary "Source Data" files.

Response to the reviewers' comments

Referee #1:

1. First, the entire study is based on one cell line that is necessarily representative of the heterogeneous breast cancer. At least one other model should be utilized to support some of the key conclusions.

(Response) We appreciate this comment. We understand that the use of other tumor cells is important for generalizing the concept suggested by the results obtained with the particular tumor cell line used in our present study. We indeed employed second and third breast cancer cell lines, such as 1d and MCF-7ras cells, to generate highly metastatic cells, 1d^{CAF1cy} and MCF-7ras^{CAF1cy} presumably expressing the E^{hi} and E/M states, by co-implantation with CAFs into recipient mice (Fig 2H and Sup. Fig S3C). Moreover, we have shown treatments with SDF-1 and TGF-1 β to boost levels of CEACAM6, E-cad and ZEB1 expressions in MCF-ras cells as well as DCIS cells (Sup Fig S6D), suggesting stromal SDF-1 and TGF- β to induce the E^{hi} and E/M states in different human breast cancer cells.

2. Second, co-injection of CAFs and cancer cells subcutaneously is a suboptimal approach. Tumor evolution can be influenced by surrounding tissue environment. Mammary fat pad injection, or better yet, intraductal injection should be employed to validate some of the key conclusions.

(Response) We disagree because our study aimed to examine effects of human CAFs, but not host murine CAFs, on tumor cells using co-implantation of human CAFs and tumor cells into mice. Significant numbers of the injected CAFs were present in growing tumor xenografts (Sup. Fig S1B, C), allowing their interactions with tumor cells in a tumor mass. If examining the interactions of host murine stromal cells with the injected breast tumor cells were the aim of this study, we would agree that mammary fat pad injection is more suitable than subcutaneous injection.

3. Third, the authors showed that knockdown of CEACAM5/6 led to E-cadherin reduction, and vice versus, indicating that these molecules mutually regulate each other. This may need to be understood better. The authors suggest that reduction of one of them destabilize the others. This is hard to imagine unless CEACAM5/6 directly interact with E-cadherin. This needs to be examined and perhaps delineation of the kinetics of the process will support this conclusion better.

(Response) We appreciate these comments. In situ PLA and immunofluorescence indicated an association between CEACAM5/6 and E-cad on DCIS^{CAF2cy}. However, immunoprecipitation failed to detect interactions between CEACAM5/6 and E-cad endogenously expressed on DCIS^{CAF2cy}, presumably due to insufficient sensitivity of this assay. We agree that detailed analyses delineating their direct interaction should be performed in a future study.

4. Fourth, Is the phenotype truly reflecting cell plasticity or genetic heterogeneity?

(Response) The reviewer is asking for experiments that go well beyond the scope of the present study. However, this question is very important. Whether genetic and/or epigenetic alterations harbored in DCIS^{CAF2cy} contribute to generation of the E^{hi} and E/M states of these cells, should be investigated at the single cell level in the future.

5. If the E^{hi} or E/M cells are sorted out and cultured in vitro separately, do they revert back to the original state? What about in vivo? This will also answer whether the changes are at the genetic level (implying some cell Darwinian selection) or at the epigenetic level.

(Response) We understand that these questions are very important, since plasticity of the E^{hi} and E/M cells has yet to be examined. We also speculate that E^{hi} and E/M are indeed interconvertible cell states in a tumor cell cluster during tumor progression. This point is now mentioned in the Discussion as an important future direction for subsequent studies.

6. Along the same lines, does the maintenance of this phenotype require the constant presence of CAFs?

(Response) In Fig 6D, we show that levels of CE ACAM5, CEACAM6, p-Src and E-cad expressions are maintained in DCIS^{CAF2cy}, passaged without CAFs up to 15 population doublings in a pure culture, indicating maintenance of the E^{hi} state in these cells. The stability of the E/M state also remains to be investigated in DCIS^{CAF2cy}.

7. Based on conventional knowledge, it is conceivable that TGF- β leads to partial EMT. but the induction of E^{hi} population is particularly interesting. It would strengthen the manuscript if there is at least one experiment looking specifically at whether the presence of this E^{hi} population is altered by inhibition of Src/TGF- β in DCIS^{CAF1cy} tumors in vivo.

(Response) We have performed the experiments requested by the reviewer. We extracted DCIS^{CAF1cy} from tumor xenografts generated by DCIS cells expressing TGF- β R2 (T β R2-), CXCR4- or GFP-shRNA admixed with CAFs. Inhibition of T β R- or CXCR4 expression by shRNA significantly attenuated lung-colonizing ability, when the resulting DCIS^{CAF1cy} was intravenously injected into recipient mice, and decreased the E^{hi} or E/M states as compared to the effect of GFP-shRNA (Fig 7E-H).

In addition, DCIS^{CAF1cy} extracted from tumor xenografts generated by DCIS cells expressing Src-shRNA admixed with CAFs, also attenuated lung-colonizing ability and decreased the E^{hi} or E/M states, as shown below (data not shown in this manuscript). These findings therefore strongly suggest that T β R or CXCR4 expression is required for generation of highly metastatic tumor cells with the E^{hi} or E/M state via Src activation in DCIS^{CAF1cy}.

8. Finally, it is difficult to understand why clinical association is only strong in Her2+ breast cancer patients - despite the fact all biology is studied in a "triple negative" model. This prompts, at the very least, the usage of a Her2+ model to bridge experimental and clinical data.

(Response) Although the level of Her2 expression in DCIS^{CAF2cy} was not shown, we did indeed observe increased Her2 protein expression in these tumor cells (data not shown). How Her2 expression is induced on parental DCIS cells by interaction with CAFs is unclear at present. However, Her2 might influence metastatic ability via associations with the E^{hi} and E/M states in DCIS^{CAF2cy}. However, several molecules (CEACAM5, CEACAM6, E-cadherin, ZEB1, Src, SDF-1/CXCR4 and TGF-β/TβR) have already been highlighted and we would not like to further increase the number of topics in this single paper. The functional roles of Her2 in CAF-induced metastasis and E^{hi} and E/M states await future study.

Referee #3:

1. The authors show that the tumorigenic properties of the DCIS cells are supported specifically by the admixed with CAFs but not with the control fibroblasts or when injected alone. However, the authors failed to demonstrate that the DCIS cells used in this manuscript are unable to activate either the control fibroblasts admixed within the NOG mice nor to activate the host fibroblasts during the time of the experiment.

(Response) In previous work (Kojima, Y., et al., Proc. Natl. Acad. Sci. USA., 107, 2009-20014, 2005), we found that when human normal mammary fibroblasts are co-injected with MCF-7ras human breast cancer cells into recipient mice, these fibroblasts can convert into myofibroblastic CAFs in a tumor mass for periods of 42-242 days during the tumor progression series. As the reviewer indicated, we did not examine whether DCIS human breast cancer cells are also competent to convert the injected normal human fibroblasts and host murine fibroblasts into CAFs in tumor xenografts for 30 days after injection. Nonetheless, CAFs boosted tumor invasion and metastasis of the co-injected DCIS cells significantly more than did either control fibroblasts or the absence of fibroblasts. Thus, we assume that potential conversion of control human fibroblasts and murine host fibroblasts into CAFs by DCIS cells would be minimal in this experimental setting.

2. While it is clear that DCIS^{CAF2cy} are more potent to colonize the lung (Sup. Fig S2D and S2E), there is any doubt that DCIS^{alone1cy}, DCIS^{alone2cy}, DCIS^{cnt1cy} and DCIS^{CAF2cy} are also highly potent to colonize the lung (Sup. Fig S2D and S2E). Thus, it seems that the metastatic capacity of the DCIS is correlated with the CAF crosstalk but not entirely dependent. Also, the ability of the parental cells to colonize the lung is missing in this set of experiments, which make difficult to reach a conclusion.

(Response) We found the metastatic property of parental DCIS cells, when subcutaneously injected into mice, to be minimal (Fig 1F and Sup. Fig S1I). Importantly, co-injection with CAFs significantly boosted the ability of DCIS cells to spontaneously metastasize into the lungs. In addition, when DCIS^{alone2cy} and DCIS^{cnt2cy} were intravenously injected into mice, lung-colonizing ability was minimal (Fig 2G, Sup. Fig S2D, S2E). In contrast, DCIS^{CAF2cy} showed greatly increased lung metastasis as compared to DCIS^{alone2cy} and DCIS^{cnt2cy}. Taken together, these findings strongly support the significance of CAFs in boosting the metastatic ability of DCIS cells.

3. I found difficult to understand why the E-cadherin level of the DCIS admixed with control fibroblasts or injected alone, present such a low level of E-cadherin staining (Figure 1B, S1E, 2E and S2A). The FACs analysis presented in Figure S1F suggest a mixed population of cells instead of two distinct cell populations within the tumor mass. Also, immunobiological analysis of E-cadherin detection presented in Figure 1A (DCIS-cnt fibro.) shows a high level of E-cadherin expression, which is not present in all the IF shown in the manuscript.

(Response) We appreciate these comments. A lower level of E-cad expression, as compared to DCIS^{CAF2cy}, was detected in cultured DCIS^{cnt2cy} by Western blot (Fig 3C, H) and immunofluorescence (Sup. Fig S3B). Consistently, E-cad staining was barely detected in tumors generated by DCIS^{alone2cy} and DCIS^{cnt2cy} (Fig 2D). In contrast, moderate E-cad staining was shown by immunohistochemistry in tumors generated by parental DCIS cells and DCIS cells admixed with control fibroblasts (Fig 1A) due to the high background level, as pointed out by the reviewer. We thus replaced these with new images, with a lower E-cad staining background, in the revised Fig. 1A.

Other major concerns need to be addressed:

4. To my point of view, a more direct evidence that the DCIS E/M cells provide leading collective cancer cell strand, would be to show, using organotypic culture assays (i.e. Fig 1E) that the DCIS E/M cells (sorted by FACs) could invade independently of CAF. Indeed, in this model, CAF are the leading cells of the invading cluster. So, if DCIS E/M cells could promote collective invasion, they should be able to replace CAF cells in this assay.

(Response) The experiments proposed by the reviewer are very important. We understand that more direct evidence is required to show the precise roles of DCIS cells with the E/M state. Isolation of DCIS cells with the E/M state should be performed in a future study.

5. To this respect, what is the mesenchymal phenotype of the DCIS E/M cells (in regards to the DCIS E^{hi})? What is the level of expression of mesenchymal-related transcription factor other than ZEB1? Do they secrete MMPs? Are they contractile?

(Response) We assume that the reviewer is asking other mesenchymal markers for confirming the E/M state. We thus performed immunohistochemistry of tumor sections generated by DCIS cells admixed with CAFs injected subcutaneously into recipient mice using anti-fibronectin and -vimentin antibodies. DCIS cells expressing E-cadherin stained positive for fibronectin and vimentin, both of which are mesenchymal markers, further indicating the E/M state in DCIS cells induced by CAFs. These new images were introduced into Sup. Fig. S1F. Accordingly, new sentences were added to the text of the revised paper.

6. Also, a clear demonstration of the importance of the two distinct cell populations in cancer growth and metastatic spreading would be obtained by independent sub-cutaneous injection of the DCIS E^{hi} and DCIS E/M after cell sorting.

(Response) We appreciate this interesting comment allowing definitive insights relevant to future work. Separation and isolation of DCIS cells with the E/M or E^{hi} state would be very interesting and we plan to carry out future experiments aimed at elucidating this issue.

7. The set of experiments obtained by multiple cell injection in vivo and isolation of the DCIS^{CAF2cy} (and controls) is an important aspect of this manuscript. It seems that cells retain the memory of the co-injection even after in vitro culture. It is stated in the discussion that epigenetic events seem not to be a crucial mechanism here. Therefore, how would you explain the long lasting effect observe using those cells?

(Response) We demonstrated that DNA methylation might be minimally involved in the CAF-induced E^{hi} state due to the slightly hypomethylated promoter region present in the CAM6 gene, but not in either the CAM5 or the E-cad gene. However, other epigenetic alterations regulating histone modifications may mediate the CAF-induced E^{hi} state. This sentence was also added to the Discussion section of the revised manuscript.

8. Since DCIS cells are barely metastatic in vivo, and their pro-metastatic capacity is induced by the CAF through the appearance of the two distinct cell populations. What is the statue of the E^{hi} and E/M populations in a more aggressive breast cancer model (i.e. human MDA-MB-231 cells or others?). Are they all E/M state?

(Response) MDA-MB-231 cells usually have undetectable levels of E-cadherin expression due to hypermethylation in the E-cadherin gene promoter region. The E/M state is induced by partial EMT in different breast tumor cell lines. However, the aggressiveness of tumor cells also depends on epi/genetic alterations harbored in these cells. Importantly, Grosse-Wilde et al., compared stemness and cell plasticity between individual tumor cell populations with the E/M state, purely epithelial state and highly mesenchymal state within whole populations of the human breast tumor HMLER cell line

(Grosse-Wilde, A et al., PLoS One. 2015, 10, e0126522). They showed that CD24- and CD44-positive tumor cells expressing the hybrid E/M states increase self-renewal and cell plasticity, indicative of tumor stemness, significantly more than do highly epithelial or mesenchymal tumor cells. In addition, the likelihood of the hybrid E/M tumor cells promoting invasion, metastasis, tumor stemness and drug resistance has been highlighted in several review articles published by authorities in the EMT field (Brabletz, T., et al., Nat Rev Cancer. 2018, 2, 128-134; Shibue, T., et al., Nat Rev Clin Oncol. 2017, 14, 611-629; Nieto, A., et al., Cell. 2016, 166, 21-45).

9. What is the level of TGF- β 1 and SDF-1 secretion in the DCIS cells? One would consider that DCIS could eventually secrete moderate level of these two cytokines, and therefore an autocrine loop could potentially take place, and therefore ruled out the hypothesis of the paper, since the DCIS alone does not show high level of DCIS E/M phenotype.

(Response) We assume that the reviewer is asking whether DCIS^{CAF2cy} maintains an autocrine signaling loop via increased levels of TGF- β 1 and SDF-1 production. To respond to the reviewer's concern, we performed real-time PCR using primers detecting TGF- β 1 and SDF-1 expressions. We observed that levels of TGF- β 1 and SDF-1 mRNA expressions were similar in DCIS^{cnt2cy} and DCIS^{CAF2cy} (right), indicating that the autocrine signaling loop is barely established in these cells. However, our preliminary results indicate that once activated, DCIS^{CAF2cy} is dependent of the TGF- β and SDF-1 signaling maintained by unknown mechanisms without constant interaction with CAFs.

10. At experimental level, DCIS^{CAF2cy} show a higher level of CAM5 (more than 100-fold of induction compare to ctrl cells) compared to CAM6 (around 40-fold) (Fig 3D), while, the TGF- β 1 and SDF-1 combo of stimulation of DCIS cells in vitro induced more CAM6 than CAM5 at low rate (7- and 1.5-fold, respectively). How the authors explain such a discrepancy, would it mean that many other mechanisms regulate CAM5 and 6 expressions in DCIS?

(Response) We appreciate these comments. As the reviewer suspected, we think that other mechanisms besides stromal TGF- β 1 and SDF-1 productions might also contribute to the inductions of CAM5 and CAM6 expressions on DCIS cells. For example, CAF-induced other cytokines, ECM remodeling, hypoxia, low nutrition, low pH and so on may be involved the inductions of these factors in a tumor mass. This *in vitro* culture model would partially mimic events in a tumor mass.

11. Finally, the concept involving CAFs and DCIS cells crosstalk for CTC cluster formation within the systemic circulation, as suggested by the schematic cartoon presented in Fig. 8F seems not to be supported enough in this work by experimental data. This concept would deserve a full manuscript of investigation by itself.

(Response) We disagree because we believe that our present experimental evidence demonstrates that CAF-produced SDF-1 and TGF- β enable the formation of invasive and metastatic tumor cell clusters with the E^{hi} and E/M states. Such tumor cell clusters could intravasate, extravasate and efficiently colonize distant organs via MET, revealing previously unappreciated findings indicative of CAF-induced highly invasive and metastatic tumor cell clusters via partial EMT. However, the biological and molecular associations between the CAF-induced E^{hi} and E/M states in tumor cells merit further detailed investigation in a future study.

Response to the editorial comments

1. In line with our general policies, we would like to display minimally processed raw data alongside the final figures for both Fig 1A and 2D, but ideally also for the other figures.

(Response) Raw data for Fig 1A and 2D were uploaded into the web submission system.

2. Please indicate in the figure legend to Fig. S1K that the same image is shown as in Fig. 1G.

(Response) We added “The image (L-lung) is also shown, as in Fig. 1G.” to the Sup. Fig S1K, L legend.

3. Note that we only have supplementary figures and tables (not appendix figures/tables) in LSA => please re-name accordingly.

(Response) The appendix figures/tables were renamed as supplementary figures and tables.

4. Please upload all figures and supplementary figures as individual files, the suppl. Figure legends can go into the main manuscript text as well.

(Response) We uploaded all figures and supplementary figures as individual files. The Sup. Figure legends were also moved to the main manuscript text.

5. Please incorporate the appendix methods into the main manuscript.

(Response) The appendix methods have now been incorporated into the main manuscript text.

6. The tables should all get provided as word docx or excel files, please

(Response) The tables were uploaded as excel files into the web submission system.

7. Note that a callout to table S7 is currently missing in the text.

(Response) “Antibodies used are listed in Table S7.” has now been added to the Immunohistochemistry and Immunofluorescence of Materials and Methods section.

8. Please increase the scale bar for the figures, they are rather small in many of them.

(Response) The small scale bars have now been enlarged.

July 4, 2019

RE: Life Science Alliance Manuscript #LSA-2019-00425-TR

Dr. Akira Orimo
Graduate School of Medicine, Juntendo University
Department of Molecular Pathogenesis
2-1-1, Hongo
Bunkyo-ku, Tokyo 113-8421
Japan

Dear Dr. Orimo,

Thank you for submitting your revised manuscript entitled "Stromal fibroblasts induce metastatic tumor cell clusters via epithelial-mesenchymal plasticity". I appreciate your response to the concerns raised by the reviewers and would be happy to publish your paper in Life Science Alliance pending the following:

- it would be good to include the data for HER2 expression in the DCIS cells (or to at least mention this in the text) to allow better linking of your experimental data to the Her2⁺ER-PR tumor status and poor outcomes in breast cancer patients
- please add a summary blurb in our submission system

A. FINAL FILES:

-- Summary blurb (enter in submission system): A short text summarizing in a single sentence the study (max. 200 characters including spaces). This text is used in conjunction with the titles of

papers, hence should be informative and complementary to the title. It should describe the context and significance of the findings for a general readership; it should be written in the present tense and refer to the work in the third person. Author names should not be mentioned.

B. MANUSCRIPT ORGANIZATION AND FORMATTING:

Sincerely,

July 10, 2019

RE: Life Science Alliance Manuscript #LSA-2019-00425-TRR

Dr. Akira Orimo
Graduate School of Medicine, Juntendo University
Department of Molecular Pathogenesis
2-1-1, Hongo
Bunkyo-ku, Tokyo 113-8421
Japan

Dear Dr. Orimo,

Thank you for submitting your Research Article entitled "Stromal fibroblasts induce metastatic tumor cell clusters via epithelial-mesenchymal plasticity". It is a pleasure to let you know that your manuscript is now accepted for publication in Life Science Alliance. Congratulations on this interesting work.

DISTRIBUTION OF MATERIALS:

Again, congratulations on a very nice paper. I hope you found the review process to be constructive and are pleased with how the manuscript was handled editorially. We look forward to future exciting submissions from your lab.

Sincerely,

Andrea Leibfried, PhD
Executive Editor
Life Science Alliance
Meyerohofstr. 1
69117 Heidelberg, Germany
t +49 6221 8891 502
e a.leibfried@life-science-alliance.org
www.life-science-alliance.org